# Combination Anticancer Therapies Using Selected Phytochemicals

**DOI:** 10.3390/molecules27175452

**Published:** 2022-08-25

**Authors:** Wamidh H. Talib, Dima Awajan, Reem Ali Hamed, Aya O. Azzam, Asma Ismail Mahmod, Intisar Hadi AL-Yasari

**Affiliations:** 1Department of Clinical Pharmacy and Therapeutic, Applied Science Private University, Amman 11931-166, Jordan; 2Department of Genetic Engineering, College of Biotechnology, Al-Qasim Green University, Babylon 964, Iraq

**Keywords:** alternative anticancer therapy, natural products, cancer, curcumin, resveratrol

## Abstract

Cancer is still one of the most widespread diseases globally, it is considered a vital health challenge worldwide and one of the main barriers to long life expectancy. Due to the potential toxicity and lack of selectivity of conventional chemotherapeutic agents, discovering alternative treatments is a top priority. Plant-derived natural products have high potential in cancer treatment due to their multiple mechanisms of action, diversity in structure, availability in nature, and relatively low toxicity. In this review, the anticancer mechanisms of the most common phytochemicals were analyzed. Furthermore, a detailed discussion of the anticancer effect of combinations consisting of natural product or natural products with chemotherapeutic drugs was provided. This review should provide a strong platform for researchers and clinicians to improve basic and clinical research in the development of alternative anticancer medicines.

## 1. Introduction

Cancer is one of the major public health problems, ranked as the second leading cause of death worldwide [1]. From a statistical perspective, 19.3 million new cases and about 10 million deaths have been reported in 2020 [2]. Cancer and its treatment have a negative impact on the economic resources and the health care system, which requires paying more attention to developing new preventive and treatment strategies with low cost and effective outcomes [2]. Additionally, other factors contributed to cancer being a global burden, including drug resistance and treatment side effects [3,4].

Since cancer is a heterogenous disease, conventional monotherapy has shown limited efficacy in the treatment and prevention [5]. In addition, several anticancer drugs have been associated with prominent undesirable adverse effects such as cardiotoxicity by doxorubicin [6], ototoxicity as a long-term side effect of cisplatin [7], and cognitive impairment by the 5-fluorouracil drug [8]. Hence, plant-derived compounds, known as phytochemicals, have been proved to be a potential approach for discovering new effective and safer anticancer agents [9]. Moreover, phytochemicals can inhibit cancer development via inducing cell apoptosis, modulating the immune response, suppressing angiogenesis factors, and targeting gene expression in cancer [10,11]. In preclinical studies, natural products in combination with chemotherapy have shown an ability to enhance anticancer activity and overcome drug resistance [12,13]. Moreover, it was found that high single doses of natural compound treatment may not be effective as using lower doses in combination anticancer treatment models [5,14]. The advantage of using a combination approach in cancer therapy is represented by targeting different pathways in a distinctively, synergistic, or additive manner [15]. In this context, when designing a combination experimental model, the expected cross-resistance and overlapping adverse effects of these compounds should be taken into account [16].

Many preclinical studies have investigated combination cancer therapies that involved natural product interventions and revealed promising results [5]. Fantini et al. [17] demonstrated how the combination treatment using different polyphenols may conquer its poor bioavailability and consequently increase their activity. On the other hand, six phytochemicals, including indol-3-carbinol, resveratrol, C-phycocyanin, isoflavone, curcumin, and quercetin, have been tested in combination against breast cancer cell lines. The results have shown a synergistic effect in inhibiting cell growth, suppressing tumor cell migration and invasion, and promoting both cell cycle arrest and apoptosis [18]. 

In this review, we aim to provide comprehensive data on the main effective phytochemicals and demonstrate their molecular mechanisms of action in combination with other plant-derived molecules or chemotherapy. Choosing these phytochemicals was based on their high potential anticancer activity and the extensive evaluation of their effect on improving chemotherapy outcomes.

## 2. Combination Therapies Based on Selected Natural Products

### 2.1. Curcumin 

Curcumin (CUR) (diferuloylmethane) is a polyphenol that is extracted from the rhizomes of the natural plant *Curcuma longa* L. (turmeric) [19,20]. It was discovered for the first time in 1870, in a pure crystalline form [20] (Figure 1). Turmeric is one of the most widely used culinary spices in India and Southeast Asian nations, and is widely used in traditional Chinese herbal medicine [21]. Curcumin exerts multiple pharmacological activities including antioxidant, anti-inflammatory, antibacterial, antiviral, and anti-cancer activity. Currently, its anticancer effect has been the most researched [22]. The main challenges facing the use of turmeric are low water solubility and bioavailability [23]. Several structural changes have been made to increase its overall anticancer efficacy and improve selective toxicity against certain cancer cells [23,24].

An in vitro study showed that turmeric with IC_50_ (31.14 ± 1.24 µM) was effective against MCF-7 cell lines in breast cancer [25]. Moreover, the IC_50_ of free CUR for 48 h was 5.63 μg/mL in Colon cancer [26]. Zargari et al. demonstrated that IC_50_ of pure turmeric after 72 h was 13.6 µM in lung cancer [27]. A toxicity study showed that curcumin exhibited limited toxicity when injected intraperitoneally in mice with LD_50_ value of 1500 mg/kg [28]. The LD_50_ of curcumin was calculated by Harishkumar et al. and was found to be 135 µg/mL in zebrafish embryos which were transferred to a 24-well cell culture plate [29].

Lower doses of curcumin were used as therapeutic doses in cancer treatment. Fetoni et al. described that curcumin was administered intraperitoneally at three different doses (100, 200, and 400 mg kg^−1^ body weight) [30]. The administration of a curcuminoid formulation (180 mg/day) as adjuvant treatment for 8 weeks to cancer patients with solid tumors significantly increased life satisfaction and reduced systemic inflammation [31].

Curcumin exhibits anti-cancer activity due to its ability to induce apoptosis, and decrease tumor growth and invasion through the suppression of a range of cellular signaling pathways [32]. Kuttikrishnan et al. demonstrated that 80 μM of curcumin-induced apoptosis in acute lymphoblastic leukemia [33]. Although extensive research has demonstrated that curcumin causes cytotoxicity in cancer cells through a variety of mechanisms. Interestingly, curcumin combined chemotherapy had increased treatment outcomes synergistically [34].

In vitro study had shown that a combination of 5 nm paclitaxel and 5 μm curcumin was highly beneficial for treating cervical cancer [35,36]. This compound enhanced paclitaxel-induced apoptosis by increasing p53 expression, activation of caspase-3, 7, 8, and 9, cleavage of poly(ADP-ribose) polymerase (PARP), and cytochrome c release, as shown by western blot analysis [35,37]. Banerjee et al. suggested that combining curcumin with standard chemotherapy might be an effective treatment strategy for individuals with prostate cancer. Moreover, reducing cytotoxicity and overcoming docetaxel-induced drug resistance. Commonly, long-term docetaxel therapy leads to drug-resistant in metastatic prostate cancer cell lines [38].

Metformin is used as a treatment for noninsulin-dependent diabetes mellitus (T2 DM) [39]. Interestingly, curcumin and metformin had a synergistic inhibition impact on prostate cancer cell line growth due to apoptotic induction [40].

Colorectal cancer has been widely treated with 5-FU alone (10 M) or in combination with other chemotherapy agents [41]. Multidrug resistance was common in individuals with colorectal cancer who were given a 5-FU-based treatment [41]. Thereby, a new therapy to overcome resistance is needed, such as combining 5-FU with curcumin in MMR-deficient human colon cancer cell lines [42]. When compared to celecoxib alone, curcumin with celecoxib inhibited colorectal cancer cell proliferation in vitro [43]. Moreover, in bladder cancer cell lines (253J-Bv and T24), co-treatment of curcumin (10 M) and cisplatin (10 M) stimulated caspase-3 and overexpressed phospho-mitogen-activated protein kinase (p-MEK) and phospho-extracellular signal-regulated kinase 1/2 (p-ERK1/2) signaling pathways [44]. Guorgui et al. found that combining curcumin (5 M) with doxorubicin (0.4 mg/mL) reduced the growth of Hodgkin lymphoma (L-540) cells by 79% [45].

In vitro and in vivo studies reported that (docetaxel/curcumin copolymers ) are strong anti-tumor candidates with tremendous promise in ovarian cancer treatment [46]. Combination of curcumin and 3-acetyl-11-keto—boswellic acid (AKBA) were shown to have antineoplastic effects in colorectal cancer in vivo. The anticancer mechanism of this combination is mediated through alteration of miRNAs and their downstream target genes involved in cell-cycle control [47]. 

Curcumin in combination with soy isoflavones inhibited the generation of inflammatory markers (prostate-specific antigen) in the LNCaP prostate cancer cell line [48]. Andrea Arena et al. found that curcumin and resveratrol were equally effective in reducing cancer cell viability in Her-2/neu-positive breast and salivary cancer cell lines. This activity was with different effects on autophagy, ROS, and PI3K/AKT/mTOR pathway activation [49]. Furthermore, this combination resulted in a higher cytotoxic impact, which was related to increased ER stress and activation of the pro-death UPR protein CHOP [49]. Curcumin and Epigallocatechin Gallate (EGCG) combination exhibited several anticancer activities [50]. When combining these two natural polyphenols, a good therapeutic effect was observed in the treatment of bladder, ovarian [51], breast [52], and prostate malignancies [53]. Furthermore, Somers-Edgar et al. had shown that a combination of EGCG (25 μM) and curcumin (3 μM) is synergistically cytotoxic toward MDA-MB-231 human breast cancer cells in vitro and decreases ERα-tumor growing in vivo [54].

In addition, 30 μM curcumin with 80 μM emodin exerted potent actions against breast cancer cell lines. Due to inducing the expression of miR-34a, the tumor growth and invasion had suppressed [55]. Another study examined the synergistic effect of curcumin and thymoquinone (TQ), on the development of MCF7 and MDA-MB-231 breast cancer cell lines [56]. Moreover, this compound and gemcitabine prevented the development, invasion, and metastasis of the pancreatic cancer orthotopic model. Those effects were due to inhibiting angiogenesis, proliferation, and downregulation of NF-κB–regulated gene products [57,58]. Aside from that, they upregulate proteins involved in apoptosis and PC cell inhibition (Bax and caspase) [57,58]. Several studies demonstrated that curcumin appears to interact with vitamin D receptors, which might explain its anti-cancer capabilities in Caco-2 human colon cancer cells [59]. Curcumin and quercetin reduced cancer cell proliferation synergistically in A375 melanoma cells. Modulation in Wnt/β-catenin signaling and apoptotic pathways are moderately responsible for the antiproliferative effects [60].

### 2.2. Resveratrol

Resveratrol (RES) (*trans*-3,4′,5-trihydroxystilbene) is a phytoalexin belonging to the stilbene class that occurs naturally. It is normally synthesized by plants in response to injury or when under attack by microorganisms including bacteria or fungi [61]. Even though 72 different plants produce resveratrol naturally, the main sources of resveratrol include wine, grapes, peanuts, pomegranate, pines, cocoa, cranberries, and dark chocolate [62]. The two principal isomers of resveratrol are *cis* and *trans* (Figure 2), and they frequently coexist. Moreover, the trans is more biologically active than the *cis* form [63]. Resveratrol may play an important role in the prevention or treatment of chronic diseases, among its effects are antioxidative, anti-inflammatory, anti-proliferative, and anti-angiogenesis properties, as well as improved cardiovascular outcomes [62,64].

Several studies were conducted to evaluate the toxicity of resveratrol. Against HeLa human cervical cancer cells, RES was active at IC_50_ value of 83.5 µM [65]. Moreover, HT-29 human colon cancer cells were inhibited by RES at IC_50_ value of 43.8 μmol/L [66]. RES displayed growth inhibitory activities against HT-29, HCT-116, and Caco-2 human colon cancer cells with IC_50_ values of 65, 25 and >100 μM, respectively [67]. Jawad et al. reported that the LD_50_ dose of resveratrol was 1.07 g/kg for males and 1.18 g/kg for females in mice after intraperitoneal administration [68].

Therapeutically, resveratrol (100 mg/kg) was intraperitoneally injected to treat lung cancer cells and the treatment resulted in tumor regression [69]. Based on the results of the previous clinical studies, the recommended dosage of resveratrol for the treatment of colon cancer is 20–120 mg daily for two weeks [70] or 0.5–1 g daily for one week [71], and 5 g daily for two weeks for patients with colorectal cancer [72].

Resveratrol has numerous chemoprotective and cancer therapy mechanisms to prevent, arrest, or reverse carcinogenesis stages. Genome instability, abnormal cell proliferation, abnormal response to signals or stimulators of programmed cell death, increased oxidative stress, overproduction of growth regulator hormones, and changes in the host immune system are among the most important cellular changes. The antioxidant, anti-inflammatory, and immunomodulatory activities also contribute, to reducing the damage caused by oxidative stress (DNA damage, protein oxidation, and lipid peroxidation) and enhancing immune oncosurveillance [73]. Resveratrol inhibits the monooxygenase cytochrome P450 isoenzyme CYP1 A1, the liver enzyme responsible for the metabolism of xenobiotics, as well as acts as a blocking agent by preventing the conversion of procarcinogen to carcinogen [74,75]. Numerous in vitro and limited in vivo studies indicate that resveratrol may augment the antitumor effects of chemotherapeutic drugs in a variety of cancers [76,77]. In addition to its anti-carcinogenic effect, resveratrol is now being studied for its potential as an adjunct in conjunction with chemotherapeutic agents to boost their efficacy and/or limit their toxicity. Using a mouse xenograft model of malignant glioma, Lin and colleagues found that resveratrol enhanced the alkylating agent temozolomide’s therapeutic efficacy by inhibiting ROS/ERK-mediated autophagy and improving apoptosis [78]. Resveratrol in 12.5 mg/kg dose has also been used to reduce chemoresistance in a mouse model of B16/DOX melanoma by inducing cell cycle disruption and apoptosis, resulting in decreased melanoma growth and increased mouse survival [79].

Malhotra and co-workers evaluated the efficacy of curcumin in combination with resveratrol in mice with benzo-a-pyrene (BP)-induced lung carcinogenesis [80]. The study demonstrated that the combination of curcumin and resveratrol enhances chemopreventive efficacy by maintaining adequate zinc levels and modulating Cox-2 and p21 [80]. Resveratrol and melatonin have also been studied in combination, NMU-induced mammary carcinogenesis was not affected by either agent alone, but when they were combined it resulted in a significant decrease in tumor incidence [81]. A combination of resveratrol, quercetin, and catechin to gefitinib can enhance its antitumor and antimetastatic effects in nude mice [82]. These studies support the possibility of using resveratrol in conjunction with chemotherapeutic drugs for cancer management.

### 2.3. Genistein

Genistein (GNT) (4,5,7-trihydroxyisoflavone) is the dominant isoflavone in soybean-enriched foods, which make up a large part of the Asian diet (Figure 3). A study found that isoflavone levels in the blood were inversely related to the risk of non-proliferative and proliferative benign fibrocystic conditions, as well as breast cancer [83]. At first, genistein was assumed to be a phytoestrogen because its structure was similar to that of estrogens and it had a small amount of estrogenic activity. The main building block of isoflavone compounds is the flavone nucleus, which is made up of two benzene rings connected by a heterocyclic pyrane ring. Due to their similar structures, it has been shown that genistein competes with 17-estradiol in ER binding tests [84].

It was discovered that genistein specifically inhibits EGFR as well as other RTKs with an IC_50_ value of 22 µM [84]. Another study showed that genistein inhibits the autophosphorylation of EGFR in vitro at an IC_50_ value of 2.6 µM [85]. The IC_50_ value of genistein against PLK1 activity was 7.9 µM while the IC_50_ values of genistein against other TKs, such as erbB2, erbB4, IGF1 receptor, insulin receptor, and PDGFR were over 4000 µM [86]. According to a study, the LD_50_ of genistein was 1150 mg/kg in mice when given intraperitoneally [87]. In HL-60 cells, genistein reduced the number of cells by causing the G2/M phase to be arrested, induced cell death through mitochondrial and ER stress-dependent pathways, and inhibited tumor characteristics in vivo. Mice were intraperitoneally injected with genistein (0, 0.2, and 0.4 mg/kg) for 28 days in an animal xenografted model and results showed tumor regression in treated animals [88].

Numerous important biological effects of genistein consumption concerning its anticancer properties have been illustrated. Even though, genistein has several health benefits, such as reducing the incidence of cardiovascular disease [89], preventing osteoporosis, and alleviating postmenopausal issues [90]. Genistein is a known inhibitor of the protein-tyrosine kinase (PTK), which may inhibit PTK-mediated signaling mechanisms to inhibit the growth of cancer cells [86]. Transgenic mice that overexpress the HER-2 gene’s tyrosine phosphorylation show delayed tumor development when genistein is given as an oral supplement, according to a study published just recently by the group Sakla et al. This shows that it may have an anti-cancer role in breast cancer chemotherapy [91]. However, it has been shown that other effects are not related to this activity [92]. It is possible that the inhibition of topoisomerase I and II [93], 5α-reductase [94] as well as protein histidine kinase [95], are all part of the mechanism by which genistein acts.

Genistein’s chemotherapeutic mechanism of action has been widely studied in a variety of cancers. Apoptosis, angiogenesis, and metastasis are all mechanisms affected by genistein. The primary molecular targets of genistein involve caspases, B-cell lymphoma 2 (Bcl-2), Bax, NF-B, PI3K/Akt, ERK1/2, mitogen-activated protein kinase (MAPK), and the Wnt/-catenin signaling pathway. Genistein has been shown to induce apoptosis in tumor cells by targeting the PPAR signaling cascade, which has surfaced as another potential therapeutic target for modulating tumor growth [96].

By modulating AMPK and COX-2, genistein with capsaicin exerted synergistic apoptotic and anti-inflammatory effects on MCF-7 human breast cancer cells [97]. It has been shown that genistein exposure for 24 h followed by 48 h of estradiol treatment resulted in the greatest apoptosis in HepG2 human liver cancer cells [98]. The anticancer effects of 5-fluorouracil in MIA PaCa-2 human pancreatic cancer cells were augmented by the addition of genistein, which increased both apoptosis and autophagy. Additional studies on animals transplanted with MIA PaCa-2 cells showed a significant decrease in tumor volume after the combination of treatments [99]. It has also been shown that genistein enhances the efficacy of photofrin-mediated photodynamic therapy to induce apoptosis in human ovarian cancer and thyroid cancer cells [51]. Activation of the general apoptotic signaling cascade required activation of caspase-8 and caspase-3 to regulate these effects [51,100]. Genistein and sulforaphane have a synergistic effect on MCF-7 and MDA-MB-231 breast cancer cells; this combination reduced cell viability, resulting in cell death, as well as cell cycle arrest in G1 phase (MCF-7 cells) and G2/M phase (MDA-MB-231 cells) [101].

### 2.4. Epigallocatechin Gallate

Many recent studies have focused on examining green tea *(Camellia sinensis)* and its polyphenolic components; one of the most interesting among these compounds is the Epigallocatechin Gallate (EGCG) (Figure 4). It is believed to have several benefits in the health sector as it has a role in various types of diseases such as cardiovascular diseases, as EGCG inhibits the NF-kappaB (NF-κB), which may be involved in developing heart failure. Additionally, EGCG inhibited myeloperoxidase (MPO) which is known to be elevated in coronary artery diseases (CAD) [102]. EGCG also has a role in metabolic diseases such as Diabetes Mellitus as it can lower the plasma glucose level and glycated hemoglobin level [102]. Furthermore, EGCG can act as an anti-oxidant due to its power in attacking reactive oxygen species [103].

To evaluate EGCG toxicity, a study demonstrated that 13 weeks of EGCG oral administration in rats was non-toxic at doses up to 500 mg/kg/day. However, oral administration of 2000 mg EGCG/kg was fatal. No toxicity was observed at an oral dose of 200 mg EGCG/kg [19]. While another study showed that the ingestion of green tea-derived supplements at a high dose (120 mg/kg) can induce toxic effects such as hepatotoxicity in rodents [104].

Additionally, EGCG has an important role in fighting cancer as it inhibits the initiation, promotion of, and progression phases in cancer cells [105]. Add to that its ability to promote apoptosis. Huang et al. found that 30 µmol/L of EGCG had induced apoptosis in MCF-7 breast cancer cell lines [106]. A study reported that the IC_50_ for EGCG when used against Eca-109 and Te-1 cancer cells was 256 and 162 μM, respectively [107]. Another article reported that the IC_50_ for EGCG which inhibited the NDPK-B activity was 150 µM [108]. Furthermore, it had been found that IC_50_ of EGCG against lung A549 cancer cells was 25 μM [109]. Additionally, reduced cell viability was reported at IC_50_ values of 14.17 μM for T47D and 193.10 μM for HFF cells [110].

Regarding toxicity, the estimated LD_50_ of EGCG when administered intradermally in rats was 1860 mg/kg [111]. Moreover, it had shown that EGCG-produced dose dependent cell death with average IC_50_ equals to 25–50 μg/mL in human B-cell lymphoma cell lines and primary NHL cells [112]. In another study, it had been shown that the IC_50_ for EGCG was 348 µM when used with A549 cells [113]. According to an in vivo study, ECGC was used in SW780 nude mice xenograft model at a concentration of 100 mg/kg, which was equivalent to a single dose of 487 mg EGCG powder for a 60-kg adult. The results have shown that ECGC successfully inhibited tumor progression in tumor-bearing mice [114]. In addition, treatment with EGCG (50 mg/kg/day, 14 days) diminished the growth of MCF-7 implanted breast tumors in athymic nude mice by 40% [115].

EGCG has an important role in fighting cancer as it inhibits the initiation, promotion, and progression phases in cancer cells [106]. Add to that its ability to promote apoptosis. Huang et al. found that 30 µmol/L of EGCG had induced apoptosis in MCF-7 breast cancer cell lines [107]. Furthermore, EGCG could be used with other anti-cancer treatments, such as natural products and chemo drugs. However, regarding the EGCG effect with natural products, Eom et al. had shown that 50 and 100 μM EGCG use along with curcumin had arrested S and G2/M cycles in PC3 prostate cancer cells [116]. In addition, EGCG improved the anti-metabolic effect of quercetin in ER-negative breast cancers, and also it decreased the viability and proliferation of MCF7 cells [117]. Furthermore, Tan et al. reported that (5, 25, and 50 μg/dL) of EGCG and thymoquinone had decreased the proliferation of PANC-1 pancreatic cancer cell lines [118]. In addition, Chen et al. demonstrated that a combination of EGCG and sulforaphane had provoked apoptosis in ovarian resistant cells in vitro, through human telomerase reverse transcriptase (hTERT) and Bcl-2 down-regulation [119]. Moreover, in vivo study reported that 30 μM EGCG combination with 15 μM resveratrol resulted in enhancing the apoptotic effect and reducing the tumor growth in head and neck cancer [120]. With chemotherapy, Wei et al. had shown that using 20–100 μM EGCG along with 5-fluorouracil (5-FU) and doxorubicin enhanced their ability in growth inhibition and also improve their ability to suppress the phosphorylation of extracellular-signal-regulated kinase (ERK) in multiple cancer cell lines [121]. La et al. also proved that 50 μM EGCG increased DLD1 colorectal cancer cell line’s sensitivity to 5-FU through the inhibition of 78-kDa glucose-regulated protein (GRP78), NF-KB, miR-155-p5, and multidrug resistance mutation 1 (MDR1) pathways [122]. Furthermore, 10 μM EGCG had enhanced cisplatin sensitivity in ovarian cancer cell lines by regulating the expression of copper and cisplatin influx transport which is well-known as copper transporter 1 (CTR1) [123]. Moreover, 100 μM EGCG improved the cytotoxic effects of cisplatin through autophagy-related pathways in an in vitro study [124]. In HeLa cervical cancer cells, 25 μM EGCG had potentiated cisplatin effects as a result of decreasing cell survival and enhancing apoptosis [125]. Though with tamoxifen, EGCG (25 mg kg^−1^) had lowered the negative estrogen receptor (ER-) in breast cancer cell lines, as it was expected to decrease protein expression of the epidermal growth factor receptor (EGFR), mammalian target of rapamycin (mTOR), and cytochrome P450 family 1 subfamily B member 1 (CYP1B) [126]. Moreover, 20 μM EGCG synergistically encouraged the effect of paclitaxel on breast cancer cells as it enhanced the phosphorylation of c-Jun N-terminal kinase (JNK) and the cell death in 4T1 cells [127]. Additionally, 20 μM EGCG had improved gefitinib resistance by inducing cell death by affecting the phosphorylation of EPK as well as the inhibition of epithelial-Mesenchymal transition (EMT) and inhibition of the phosphatidylinositol-3-kinase (PI3K)/mTOR pathway in non-small cell lung cancer (NSCLC) cell lines [128]. Besides this, EGCG had improved the effect of erlotinib in head and neck cancer in vitro. As it enhanced the apoptosis through the regulation of Bcl-2-like protein 11 (BIM) and B-cell lymphoma 2 (Bcl-2) [129].

### 2.5. Allicin

Allicin (ALN) or diallyl thiosulfinate (Figure 5) is one of the well-known organosulfur compounds that are found in garlic (*Allium sativum* L.). It can be generated by the cleavage or cutting of the garlic clove which in return activates the allinase enzyme resulting in the hydrolysis of non-proteinogenic amino acid S-allyl cysteine sulfoxide or known as (alliin) and mainly producing allicin [130].

Regarding allicin cytotoxicity, a study reported that the exposure to 12 µg/mL of allicin for 24 h, produced cytotoxic effect on MGC-803 and SGC-7901 cancer cells, including cellular membrane breakage [131]. While a study reported that allicin prevented proliferation of human mammary (MCF-7), endometrial (Ishikawa), and colon (HT-29) cancer cells at 50% inhibitory concentration equals to 10–25 μM [132]. Moreover, another study stated that when allicin used against MGC-803 and SGC-7901 cancer cells, the IC_50_ was 6.4 µg/mL, 7.3 µg/mL, respectively [131], while the LD_50_ of allicin was 120 mg/kg subcutaneous injection and 60 mg/kg intravenous injection in mice [133]. An in vivo study on bladder cancer has shown that allicin can delay the beginning of tumors following subcutaneous injection at a concentration of 12.5 mg and 25 mg [134].

Allicin has many activities, such as anti-oxidant [135] and antimicrobial [136]. Furthermore, it has a role in neuroinflammatory, and cardiovascular diseases [137], and an important role in combating cancer [138] due to its multiple mechanisms such as inducing apoptosis, inhibiting tumor growth, and preventing tumor angiogenesis [139]. For instant, 30 and 60 µg/mL of allicin induced apoptosis in U251 human glioma cells [140].

Many researchers had also studied the effects of allicin in combination therapies with other anti-cancer treatments including anti-cancer drugs and other plants. In one study, a mixture of allicin (ALN) and thymoquinone (TQ) has an excellent effect on anti-oxidant parameters in prostate and colon cancer cells [141]. Wamidh Talib reported that consumption of garlic (allicin rich extract) with lemon aqueous extract had decreased angiogenesis and induced apoptosis in breast cancer cells [142]. Moreover, Sarkhani et al. revealed that a mixture of allicin and methylsulfonylmethane had enhanced apoptosis because it increased the expression of caspase-3 mRNA expression in CD44± breast cancer cells [143].

On the other hand, allicin with antineoplastic drugs showed promising results. For example, allicin with cisplatin had shown many beneficial effects whether in fighting cancer or other helpful aspects. Pandey et al. demonstrated that using a low dose of allicin with cisplatin can potentiate the inhibitory activity of cisplatin and overcome the resistance of cisplatin. This is achieved by affecting hypoxia, which is known as a major mediator in cisplatin resistance, as allicin along with cisplatin had boosted the apoptosis in a ROS pathway in both normoxia and hypoxia [144]. Tigu et al. have reported that there was a synergistic effect against lung and colorectal cancer cells when allicin was used along with 5-FU [145]. Furthermore, allicin improved 5-FU resistance in gastric cancer cells by lowering the expression of Wnt Family Member 5A gene (WNT5A), CD44 receptor, MDR1, p-glycoprotein (p-gp) [146]. Fayin also reported that allicin had improved the apoptosis effect of 5-FU in MEC-1 cells [147]. Moreover, Xi et al. revealed that a mixture of allicin and Adriamycin had inhibited the proliferation and induced apoptosis in gastric cancer [148]. Additionally, allicin had improved the effectiveness of tamoxifen in the existence or lacking 17-b estradiol [149].

Moreover, Wu et al. revealed that allicin had protected the auditory hair cells, and spiral ganglion neurons from the apoptosis that is triggered by cisplatin [150], such result supports the fact that allicin can help in protecting from vestibular dysfunction [151]. In addition to this, a mixture of allicin and ascorbic acid alongside cisplatin displayed a neuroprotective effect against cisplatin due to allicin anti-oxidant and anti-inflammatory effects [152]. While with doxorubicin, allicin had improved the cardio-toxic effects of this anti-cancer drug by inhibiting oxidative stress, and inflammation [153]. Moreover, allicin with 5-FU had improved chemotherapy sensitivity in hepatic cancer cells due to induction of apoptosis by ROS-mediated mitochondrial pathways [154].

### 2.6. Thymoquinone

Thymoquinone (TQ) (2-Isopropyl-5-methylbenzo-1, 4-quinone) is a monoterpenoid compound [155] (Figure 6). It is extracted from the volatile and fixed oil of *Nigella sativa* (black seed) [156]. TQ is therapeutically active as an anti-microbial, anti-inflammatory, hypoglycemic, antiparasitic, antihypertensive, and anticancer agent [157].

TQ showed a significant antitumor effect on various types of cancer such as breast cancer [158], prostate cancer [159], gastric cancer [160], and bladder cancer [161]. Interestingly, TQ IC_50_ value was found to be 46 μM in a hepatocellular carcinoma cell line [162]. TQ is considered a safe natural product as its LD_50_ values for oral administration are 300–2400 mg/kg in mice and 250–794 mg/kg in rats [163]. While its therapeutic dose was about 10 mg/kg/intraperitoneally in mice [164].

Numerous studies demonstrated TQ anticancer mechanisms. Generally, it exerts its antitumor activity by modulating epigenetic machinery, altering gene expression of non-coding RNAs [165]. Moreover, via affecting several biological pathways that are implicated in apoptosis, proliferation, cell cycle regulation, and cancer metastasis [166]. In bladder cancer cell lines, 40 mmol/L of TQ stimulated apoptosis via ER-mediated mitochondrial apoptotic pathway [161].

TQ combination with various chemotherapeutic agents had enhanced the anticancer activity of them. For example, 46 μM TQ along with 64.5 μM resveratrol is considered a novel therapeutic strategy in the HCC cell line. Their combination resulted in significant cell inhibition and increased caspase-3 to induce apoptosis [162]. In an in vivo study, (20 mg·kg^−1^) of oral TQ improved the effectiveness of cisplatin in HCC treatment via controlling the GRP78/CHOP/caspase-3 pathway [167]. Furthermore, in breast cancer treatment, a combination of TQ and paclitaxel remarkably increased the rate of apoptotic/necrotic cell death in T47D cells, and induced autophagy in MCF-7 cells [168]. In vitro and in vivo models study reported that 10 μM TQ with 50 nM doxorubicin combination, enhanced cell death in adult T-cell leukemia. Thus, it increased ROS and resulted in disruption of the mitochondrial membrane [169]. A triple combination of (20 mg/kg) TQ, (15 mg/kg) pentoxifylline, and (7.5 mg/kg) cisplatin in mice, enhanced the chemotherapeutic activity of cisplatin by Notch pathway suppression [170]. A synergistic antitumor effect was detected between (10 mg/kg)TQ and (1 mg/kg) melatonin leading to minimizing the tumor size with a 60% percentage cure according to an in vivo study [171]. Similar to many chemotherapeutic agents, TQ can significantly enhance the effect of other natural products. TQ and royal jelly (RJ) together enhanced the anticancer activity of both against MDA-MB-231 breast cancer cells [172]. Moreover, in breast adenocarcinoma, a combination of (50 and 100 µM) TQ and (450 µM) ferulic acid required the use of lower doses of both to suppress the proliferation of cultured MDA-MB 231cells [173]. Additionally, TQ and quercetin potentiate apoptosis in NSCLC cell lines via the Bax/Bcl2 cascade [174]. A significant improvement in anticancer activity was examined when combined TQ with piperine (PIP) in EMT6/P cells injected in Balb/C mice. The combination treatment of (25 mg/kg/day of PIP and 10 mg/kg/day of TQ for 14 days) lead to a remarkable dropping in tumor size with a 60% of cure [175].

### 2.7. Piperine

It is most commonly found in the fruits and roots of *Piper nigrum* L. (black pepper) and *Piper longum* L. (long pepper) in the Piperaceae family as piperine (1-Piperoylpiperidine) [176] (Figure 7).

In vitro and in vivo anticancer effects of Piper nigrum extracts on colorectal cancer cells (HCT-116) and lung cancer cells (A549) were with IC_50_: HCT-116: 165 µM A549: 135 [177]. Another study by Gunasekaran et al. showed that IC_50_ was 75 µM (24 h) 30 µM (48 h) in Hepatocellular cancer [178]. Moreover, in leukemia IC_50_ was 25 µM (24 h) [179]. Regarding toxicity, after intravenous administration piperine LD_50_ was 15.1 mg per kg for adult mice [180]. In BALB/C mice implanted with mouse mammary EMT6/P cancer cells, the intraperitoneal treatment of piperine (25 mg/kg/day for 14 days) considerably reduced the tumor size [181]. In breast cancer, female BALB/C bearing 4T1 cell were treated with 2.5 or 5 mg/kg piperine every 3 days and tumor regression was reported [182]. Piperine inhibited lung metastasis of melanoma cells after its intraperitoneal injection at a concentration of 200 µmol/kg [183]. It also inhibits cell proliferation in prostate cancer cells implanted in nude mice at a therapeutic dose of 100 mg/kg/day (intraperitoneal) [184].

Piperine (PIP) activates apoptotic signaling cascades, inhibits cell proliferation, arrests the cell cycle, alters redox homeostasis, modulates ER stress and autophagy, inhibits angiogenesis, induces detoxification enzymes, and sensitizes tumors to radiotherapy and chemotherapy [185]. These mechanisms of action can help to prevent cancer. It can activate both intrinsic and extrinsic apoptotic pathways at the molecular level. Piperine suppressed mouse 4T1 breast tumor growth and metastasis [182]. Administration of piperine activated caspase 3-mediated intrinsic apoptosis in 4T1 cells and induced G2/M phase cell cycle arrest [182]. In another study, piperine reduced tumor growth in nude mice xenografted with androgen-dependent (PC3) and independent (LNCaP, DU145) prostate cancer cells [184]. It also inhibits prostate cancer cell growth by reducing phosphorylated STAT-3 and NF-B [184].

A variety of cell and tissue-specific and dose-dependent effects of piperine-mediated redox change cellular physiology. It can either enhance cell survival or commit the cell to death, depending on the situation. Oxidative stress-induced cell damage can be prevented by quenching ROS and other reactive metabolic intermediates, such as free radicals, with piperine [186,187]. A variety of protein regulators and checkpoints have been linked to the ability of piperine to halt the progression of cancer cells at various points in the cell cycle. Piperine in 100–200 µM concentration led to apoptosis and G1 phase cell cycle arrest in melanoma cells via activation of Checkpoint Kinase-1 [188].

In vitro, piperine demonstrates a synergistic anticancer effect when combined with paclitaxel on the MCF-7 cell line [189]. Another study indicates that combinations of piperine, hesperidin, and bee venom enhance the anti-cancer effects of tamoxifen in MCF7 and T47D cell lines [190]. In addition, the combination of piperine and doxorubicin inhibited tumor growth in BALB/C mice subcutaneously injected with MDA-MB-231 cells in vitro more effectively than either agent alone [191]. Piperine inhibits hepatic CYP3A4 activity in vivo, correlating with an increase in docetaxel’s AUC, half-life, and maximum plasma concentration. In addition, the synergistic administration of piperine and docetaxel significantly improved the antitumor efficacy of docetaxel in a castration-resistant human prostate cancer animal model [192]. Additionally, a study using in vitro and in vivo models, showed that the piperine and thymoquinone combination exerted a synergistic inhibition in breast cancer. This mainly was achieved by inhibition of angiogenesis, induction of apoptosis, and shifting toward T helper1 immune response [181].

### 2.8. Emodin

Emodin (EMD) is a natural anthraquinone derivative. Chemically it is (1,3,8-trihydroxy-6-methyl-anthraquinone) [193,194] (Figure 8). This phytochemical has been extracted from different Chinese medicinal herbs including *Radix rhizoma Rhei*, *Aloe vera*, *Polygonum multiflorum*, *Giant knotweed*, *Rheum palmatum*, and *Polygonum cuspidatum* [194,195,196]. Moreover, it can be found in the bark and roots of many other different plants, molds, and lichens [197].

Recently, emodin earned attention due to its diverse activity. It displays antibacterial [198], anti-inflammatory, antioxidant, antiallergic, antihypertensive, antidiabetic, neuroprotective, and hepatoprotective properties [199,200,201,202,203]. It may be used as a photosensitizing agent in photodynamic therapy [204]. In addition, it prevents immunosuppression and exhibits anticancer activity [205,206]. Emodin has shown its antitumor activity against colon cancer, breast cancer, non-small-cell lung cancer, ovarian cancer, prostate cancer, pancreatic cancer, leukemia, and hepatocellular carcinoma (HCC) [207,208].

Narender et al. reported that emodin cytotoxicity was 3.5 μM in HepG2 cell line [209]. Regarding emodin toxicity, Luo tao et al. found that 100, 200 and 400 μM of emodin resulted in reproductive toxicity in humans when applied to ejaculated human sperm [107], whereas its therapeutic dose in athymic nude mice injected with MDA-MB-231 breast cancer cells was 40 mg/kg after intraperitoneal injection [210].

Emodin displays its anticancer effect on different cell lines with different mechanisms. Generally, emodin exerts its anti-tumor activity by inducing mitochondrial apoptosis and inhibiting pathways that promote proliferation, inflammation, angiogenesis, and tumorigenesis [211]. In colon cancer (CC), emodin regulated the localization and expression of Bcl-2 family proteins by regulating PI3K/AKT, MAPK/JNK, STAT, and NF-κβ molecular signaling pathways [212]. Moreover, it inhibited the migration and invasion of CC cells by downregulating epithelial-mesenchymal transition via the Wnt/β-catenin signaling pathway [213]. More interestingly, treatment with emodin led to mitochondrial dysfunction, reactive oxygen species accumulation, and induced apoptosis in (CC) cells via induction of autophagy [214]. Furthermore, in HCT116 human (CC) cells, 10–50 µM emodin-induced apoptosis inhibited proliferation, suppressed the expression of fatty acid synthase (FASN), inhibited intracellular FASN activity, and fatty acid biogenesis. Needless to say, (FASN) is an important factor in the development of colon carcinoma [215].

Interestingly, emodin’s benefits are not limited to natural products alone, but again, it can improve the anticancer effect of several chemotherapeutic agents. Emodin’s combination with sorafenib resulted in improving the anti-cancer effect of sorafenib in HCC cells. Furthermore, this combination synergistically increased apoptotic cells and cell cycle arrest in the G1 phase using concentrations of 20 μM emodin and 2 μM sorafenib [207]. Moreover, a combination with EGFR inhibitor afatinib resulted in a higher rate of inhibiting cell proliferation in pancreatic cancer in concentrations ranging between 30, 60 and 90 μM of emodin [216]. Furthermore, the inhibition of the growth effect of cisplatin was remarkably improved by emodin in lung adenocarcinoma A549/DDP cells [217]. In addition, in endometrial cancer cells, emodin and cisplatin combination inhibited the expression of drug-resistant genes by decreasing the reactive oxygen species (ROS) levels. Consequently, resulting in increasing chemosensitivity [218]. Shuai Peng et al. demonstrated that emodin (5 µM) enhanced H460 and A549 cell sensitivity to cisplatin through P-glycoprotein downregulation in non-small cell lung cancer (NSCLC) [219]. More and more, emodin with a concentration between (5, 10, 20, and 40 μM) enhanced the anticancer effect of paclitaxel by inhibiting the proliferation of A549 cells in NSCLC [212]. In pancreatic cancer, emodin (40 μM) inhibited IKKβ/NF-κB signaling pathway and reverses gemcitabine resistance [213]. Generally, a combination of natural products has shown promising results in treating disease, either as synergistic or as an additive effect [5]. In breast cancer, a combination of emodin (10 μM) and berberine (10 and 5 μM) synergistically repealed the SIK3/mTOR pathway. As a result, the aerobic glycolysis and cell growth were suppressed leading eventually to inducing apoptosis [220].

### 2.9. Parthenolide

Parthenolide (PTL) is a germacrene sesquiterpene lactone [215]. Chemically, it consists of an α-methylene-γ-lactone ring and epoxide group, which are responsible for interacting with nucleophilic sites of biological molecules [221] (Figure 9). PTL is extracted from different plants of the Asteraceae family [222] and is the main constituent of the feverfew medicinal plant, Tanacetum parthenium [223]. Generally, it possesses diverse biological activity extending from antibacterial, anti-inflammatory, and phytotoxic to antitumor activity [224].

PTL IC_50_ values were 9.54 and 8.42 μM against MCF-7 and SiHa cells, respectively [225]. Regarding to a study, PTL showed LD_50_ at 200 mg/kg, when administered orally [226]. On the other hand, 10 mg·kg^−1^·day^−1^ of PTL administered intraperitoneally, was therapeutically effective as anticancer agent in mice injected with U87MG cells [227].

PTL has been reported as an anticancer agent using different mechanisms. Mostly, by inhibiting the nuclear transcription factor-kappa (NF-κB) signaling pathway and cell growth [221]. Add to that its ability to induce apoptosis and G0/G1 cell cycle arrest [223]. PTL stimulated apoptosis in 50–200 µmol/L concentration in human uveal melanoma cells [228]. Therefore, it is active against different types of cancer including colorectal cancer [222], breast cancer [229], and lung cancer [230].

A PTL (9 and 15 µM) combination with Epirubicin (EPR) (2.5 and 3.5 µM), which is an anthracycline doxorubicin analog, led to improving cytotoxicity and apoptosis in MDA-MB-468 breast cancer cells. Thus, the dose of EPR could be reduced and the undesirable side effects will be preventable [221].

Furthermore, in vitro study considered PTL as a potent agent at a concentration of 1 μg/mL, as it enhanced the effectiveness of arsenic trioxide (2 µM) in the treatment of adult T-cell leukemia/lymphoma [231]. Se-lim Kim et al. demonstrated that PTL 10 μM combination with balsalazide improved the anticancer activity via blocking NF-κB activation and therefore prevented colon carcinogenesis from long-lasting inflammation [221]. In addition, PTL sensitized colorectal cancer cells resistant to tumor necrosis factor-related apoptosis-inducing ligand. That was achieved by increasing the surface expression of death receptor 5 proteins, upregulating the expression of proteins elaborate in the mitochondrial apoptotic pathway, and lastly increasing caspase activation [223]. Se-lim Kim et al. demonstrated that using (5 or 10 μmol/L) PTL combination with 20 mmol/L balsalazide in vitro and in vivo improved the anticancer activity via blocking NF-κB activation. Therefore preventing colon carcinogenesis from long-lasting inflammation [232]. Recently, a combination of natural products is of interest, because they are safe, inexpensive, and effective. For instance, PTL (1.5 μg/mL) and different concentrations of ginsenoside compound K have acted synergistically as antineoplastic agents with minimizing adverse effects both in vitro and in vivo [233]. Once more, an interesting in vitro and in vivo study showed that a cocktail combination of PTL, betulinic acid, honokiol, and ginsenoside Rh2 displayed a synergistic activity in liposome systems for lung cancer treatment [234].

### 2.10. Luteolin

Luteolin (LTN) (2-[3,4-dihydroxyphenyl]-5,7-dihydroxy-4-chromenone) [235] (Figure 10) is a flavonoid that can be found in fruits and vegetables, such as parsley, sweet bell peppers, celery, onion leaves, chrysanthemum flowers, carrots, and broccoli [229]. Several studies have shown that LTN owns diverse biological activities. For instance, it acts as a neuroprotective [236], anti-diabetic, antioxidant, anti-microbial, anti-allergic, anti-inflammatory, chemopreventive, and chemotherapeutic agent [237].

Seo et al. demonstrated that LTN IC_50_ was 9.8 μM against PC-3 prostate cancer cell lines [238]. According to a study, luteolin LD_50_ was 150 mg/kg when delivered through nasogastric intubation in rats [239]. While 40 mg/kg of LTN was able to suppress the Nrf2 signaling pathway and cancer development in vivo [240]. Luteolin displays its antineoplastic activity in the forms of diverse mechanisms including hampering the activity of epigenetic targets, such as DNA methyltransferases [241], inducing autophagy, cell apoptosis, and inhibit migration and invasion [242]. A study demonstrated that 10–30 µM of LTN stimulated apoptosis and autophagy in glioma [243].

Interestingly, luteolin showed a synergistic anticancer effect with 5-fluorouracil on HepG2 and Bel7402 cells in human hepatocellular carcinoma. This effect was achieved using various dose ratios (luteolin:5-fluorouracil = 10:1, 20:1, 40:1) [244]. In drug-resistant ovarian cancer, 10, 50, and 100 μM of LTN significantly sensitized the antineoplastic effect of 2 μg/mL cisplatin. Thus initiating apoptosis and inhibiting cell invasion and migration both in vitro and in vivo [245].

A study revealed that a combination of luteolin and quercetin in (50–1000 mg/mL) concentration, synergistically improved the antitumor effect of 5-Fluorouracil (5-FU) in HT 29 cells. Consequently, it minimizes the unwanted toxic effects of 5-FU in colorectal cancer treatment [246]. Furthermore, in vitro study reported that 10 and 20 μM luteolin and 20 and 40 μM quercetin inhibited the invasion and migration of squamous carcinoma decreasing Src/Stat3/S100A7 signaling [247]. Moreover, (10, 20, and 40 μM) of luteolin and quercetin together caused a reduction in ubiquitin E2S expression led eventually to metastatic inhibition of A431-III cervical cancer cells [248]. Furthermore, when 100 or 140 mg/mL of luteolin was combined with hesperidin, an enhancement in their anticancer activity was achieved. That is due to the declining cell viability and suppression of cell cycle progression in MCF-7 cells [249]. Similarly, 20 µM luteolin and 50 µM silibinin worked synergistically together, especially in preventing cell proliferation, migration, and invasion in human glioblastoma SNB19 and GSC cells, as well as in the drug-resistant glioblastoma stem cells [250].

### 2.11. Quercetin

Quercetin (QUR) is one of the most well know flavonoids that are found in many types of fruits and vegetables; it is a flavonol that is one of the six types of flavonoids (Figure 11). Quercetin is aglycone in nature thus mainly it is not soluble in cold water, poorly soluble in hot water, and fairly soluble in lipids and alcohol as a result it is mainly attached to a glycosyl group using sugar as glucose, rhamnose, or rutinose to improve the quercetin solubility [251].

According to its cytotoxicity, a study stated that the IC_50_ of quercetin was 30 μM, which was calculated in vitro by the MTT colorimetric assay [252]. Quercetin LD_50_ was 97 mg/kg when administered subcutaneously, while its LD_50_ after intravenous administration was about 18 mg/kg in a mouse model [28]. When quercetin used in vivo at concentration of 100 and 200 mg/kg in mice bearing CT-26 and MCF-7 tumors, it showed significant higher survival rate compared to control [253]. Another study reported that administration of 10 mg/kg of quercetin intraperitoneally had inhibited cell proliferation in HepG2 tumor-bearing BALB/C/nu mice [254].

Quercetin has been utilized in different areas due to its different mechanisms such as antioxidant [241], antimicrobial [242], and anti-inflammatory [255]. It also has a great role in cancer, as it controls many factors in the cancer activity such as apoptotic proteins, cell cycle, and angiogenesis [256]. As an example, 25, 50 µM of quercetin induced apoptosis and DNA fragmentation in HeLa cervical cancer cells [257]. For these reasons, many researchers studied the final effects when quercetin had used with natural products and other anti-cancer drugs. Quercetin works synergistically with curcumin in the triple-negative breast cancer cell line by altering the BRCA1 deficiency and therefore augmenting the activity of anti-cancer drugs [258]. Moreover, quercetin and curcumin enhanced the apoptotic effect of K562 cells in chronic myeloid leukemia due to the increase in ROS and impairment of the mitochondrial membrane potential [259]. Using resveratrol with quercitin can cause DNA injury, cell growth inhibition, stimulation of apoptosis in oral cancer cell lines. It promoted apotosis via downregulation of Histone deacetylase (HDAC)1, HDAC3, and HDAC8 [260]. Moreover, a promising nanostructured lipid carrier (NLC) gel of quercetin and resveratrol had shown an improvement in the deposition of these two drugs to the epidermal layer in skin cancer cells [261]. Furthermore, combining thymoquinone with quercitin enriched the apoptosis in non-small lung cancer cell lines due to the modulation of anti-apoptotic protein Bcl2 and the initiation of proapoptotic Bax [174]. In addition, it was found that using luteolin with quercitin can prevent the invasion of cervical cancer cells as a result of a lowering in ubiquitin E2S ligase (UBE2S) [248]. With chemotherapy, quercetin potentiates the effect of cisplatin in cervical cancer cells due to the induction of apoptosis as a result of declining Matrix Metallopeptidase 2 (MMP2), Methyltransferase 3, N6-Adenosine-Methyltransferase Complex Catalytic Subunit (METTL3), P-Gp and ezrin production [262]. Using quercetin with 5-FU increased the sensitivity of MCF-7 breast cancer cells toward 5-FU [263]. On the other hand, combining quercetin with tamoxifen improved its effect on resistant breast cancer cells [264]. Moreover, quercetin had improved doxorubicin’s accumulation in breast cancer cells by downregulating the expression of efflux receptors, including breast Cancer Resistant Protein (BCRP), P-gp, and multidrug resistance protein 1 (MRP). It also lowered the side effects of doxorubicin [265]. In addition, nano-querectin had improved the cytotoxicity of doxorubicin in MCF-7 breast cancer cells [266]. Fang et al. reported that mesoporous silica nano-particles loaded with quercetin had improved the efficacy of doxorubicin treatment in gastric cancer cell lines [267]. In hepatocellular carcinoma (HCC), quercetin potentiated the growth suppression effect of cisplatin in HepG2 cells [268]. In addition, Zhu et al. reported that quercetin potentiates the effect of vincristine when delivered as nanocarriers in lymphoma in vitro and in vivo model [269]. It is worth mentioning that adding quercetin with paclitaxel therapy has improved the anticancer effect in prostate cancer both in vitro and in vivo, through triggering ROS production, induction of apoptosis, preventing cell migration and stimulating cell arrest in the G2/M phase [270]. Moreover, QUR and paclitaxel had enhanced the multi-drug resistance in breast cancer MCF-7/ADR cell lines and in vivo by decreasing P-gp expression and inhibiting of the cellular paclitaxel reflux [271]. In addition, Huang et al. revealed that nanoparticles loaded with quercetin had improved tumor targeting and radiotherapy treatment in 4T1 cells and in mice [272]. In combination with other chemo-drug, Li et al. reported that using quercetin with cisplatin had improved the apoptosis in oral squamous cell carcinoma (OSCC) cell lines and mice. This is due to the inhibition of NF-κB thus downregulating of X-linked inhibitor of apoptosis protein (xIAP) [273]. Furthermore, it increased the growth inhibition of cisplatin in breast cancer in mice [274]. Additionally, Gonzalezet et al. revealed that quercetin had improved the nephrotoxicity that accompanied cisplatin in rats [275]. Moreover, it improved oral mucositis which is induced by 5-FU in mice [263]. In addition, it offered protection to damaged peripheral nerves associated with vincristine use due to quercetin’s role in decreasing the oxidative stress, inflammation, stress and neuronal cell damage in rats [269].

### 2.12. Anthocyanins

Anthocyanins (ACN) are water-soluble flavonoids seen as pigments in the dark color of fruits and vegetables such as berries, pomegranates, berries, and rice [276]. They give different colors depending on their pH, they may appear red, purple, blue, or black. Their fundamental structural part is 2-phenylchromenylium (flavylium) [277] (Figure 12).

They are active in a variety of health conditions such as cardiovascular [278], neurological [279], and metabolic diseases [280]. Moreover, anthocyanins have an active role in cancer management due to their basic specification as anti-oxidants, anti-inflammatory, anti-invasion, and anti-metastatic [281].

A study revealed that 146–2199 mg/100 g of anthocyanin exerted a good antioxidant as well as anticancer activity [282]. Based on numerous studies, anthocyanins toxicity is considered low. For instance, a study revealed no significant effect upon 90 days intake of 0–1000 mg/kg/day anthocyanin in ovariectomized rats [283]. Furthermore, animal studies had not recognized any lethal effects regarding anthocyanins (from blueberries, currants, and/or elderberries). Moreover, the IC_50_ value for anthocyanin at 24 h after treating DU-145 cells was 60–90 µM [284]. In this context, the LD_50_ values for highly purified extract of Vaccinium myrtillus berries containing 36% anthocyanosides were over 2000 mg/kg in mouse and in rats without any toxic symptoms [285]. Moreover, in BALB/C nude mice bearing ErbB2 positive breast cancer, the oral administration of black rice anthocyanins (150 mg/kg/day) decreased transplanted tumor development, hindered pulmonary metastasis, and reduced lung tumor nodules [286].

Due to the valuable activity of the anthocyanins, many researchers investigated the outcomes when they are combined with other anti-cancer therapies including drugs and natural products. For instance, Yin et al. reported that cyanidin 3 glucoside chloride acts along with luteolin by increasing apoptosis and inhibiting the proliferation of breast and colon cancer cell lines [287]. Regarding combining anthocyanins with other chemotheraputic agents, Li et al. revealed that a combination of 5-FU and 50 μg/mL blackberries anthocyanins decreased the proliferation and migration of SW480 cells in colorectal cancer [288]. Paramanantham et al. stated that 400 µg/mL of anthocyanins isolated from *Coignetiae pulliat* had advanced the sensitivity of cisplatin in MCF-7 breast cancer cells resulting from the impairment of Akt and NF-κB activation [289]. Furthermore, an anthocyanin called cyanidin had been noticed to decrease the cardiotoxicity that is associated with cisplatin in 40–80 µM doses through preventing ROS-mediated apoptosis in H9c2 cells [290]. Pepe et al. also reported the cardio-protective effect of *Citrus sinensis* and *Vitis vinifera* anthocyanins with doxorubicin in vitro at a range between 1–25 µg/mL [291]. In addition, anthocyanins extracted from *Oryza sativa* L. and 5-FU improved the oral mucositis in vitro and in vivo using 500 mg/kg and 1000 mg/kg concentrations. This is by the activation of Nuclear Factor-κB which resulted in anti-inflammatory effects [292]. Anthocyanin from purple sweet potato had decreased doxorubicin cardiac toxicity using different concentrations (100, 200, and 400 μg/mL) according to in vitro and in vivo study. The previously mentioned effect was due to the decrease in inflammatory factors, such as nitric oxide and TNF-α, also due to the decline in creatine kinase, trimethylamine oxide, and lactic dehydrogenase triggered by myocardial damage [293]. Moreover, with 20 μg/mL trastuzumab, 1 μg/mL anthocyanins cyanidin 3 glucoside proved to show a synergistic effect in vitro and in vivo. As it had been noticed to decrease human epidermal growth factor receptor 2 (HER2) and improved the trastuzumab apoptotic effect in HER2-positive breast cancer [294]. Moreover, using 0.003–50 µM in a 100 µL of cyanidin 3 glucoside has shown to overcome trastuzumab-resistant in breast cancer cell line and mice xenograft model. The previous activity was due to decreasing the HER2, AKT, and MAPK activities [295]. Furthermore, Qi et al. had noticed that (200 and 400 mg/kg) anthocyanin from the fruits of Panax ginseng had improved the nephrotoxicity in mice, which is associated with cisplatin usage due to their anti-inflammatory and anti-oxidant influences [296]. Moreover, Gomes et al. reported the same nephroprotective effect with blackberries juice anthocyanins in mice but with a 10 mL/kg concentration [297]. Furthermore, Shi et al. had shown that the blueberry anthocyanins in a dose of 20 and 80 mg/kg/day for 7 days, had improved the liver damage in rats. Generally, liver damage is associated with cyclophosphamide usage due to the reduction of inflammation and apoptosis [298].

## 3. Conclusions

A combination of plant-derived natural products with other anti-cancer therapies showed a significant improvement in cancer management. Higher efficiency and lower toxicity were reported when combining these natural products with standard anticancer agents or other natural products. Curcumin, thymoquinone, and quercetin were extensively tested in combination anticancer therapies. Other plant-derived natural products were less tested. This could be due to several factors including: availability of the natural product, solubility, lack of clear mechanisms of action, and the cost of purchasing some natural products. Breast cancer was the most studied cancer in combination therapies in vivo and in vitro. Due to the limitation of current anti-cancer treatments such as toxicity, low solubility, low bioavailability, and resistance, combinations based on natural products is a promising strategy to develop more effective and less toxic treatments. Further studies are needed to design effective combinations of natural products that can augment conventional treatments. More studies are also needed to test complex combinations containing more than 2 natural products. Furthermore, the spectrum of activity of these combinations should be further expanded as many of the products were tested on limited cancer types. Figure 13 summarizes the main combination therapy of the natural compounds with other plant-derived compounds as well as chemotherapies. Table 1 shows the tested combination experimental design of natural compounds with other natural products and the outcomes of these studies. Table 2 demonstrates the main studies that included natural compounds in combination with chemotherapy.

## Figures and Tables

**Figure 1 molecules-27-05452-f001:**
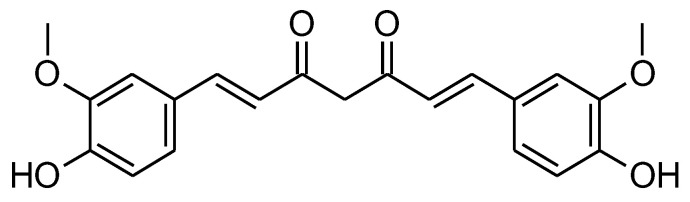
Chemical structure of curcumin.

**Figure 2 molecules-27-05452-f002:**
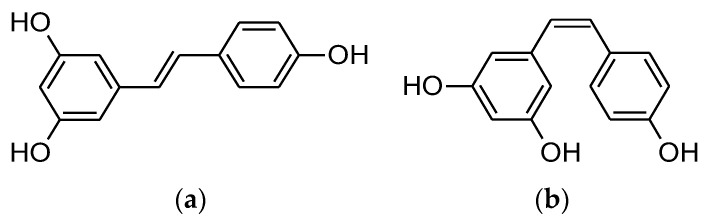
Chemical structure of resveratrol. (**a**) *Trans*-resveratrol and (**b**) *Cis*-resveratrol.

**Figure 3 molecules-27-05452-f003:**
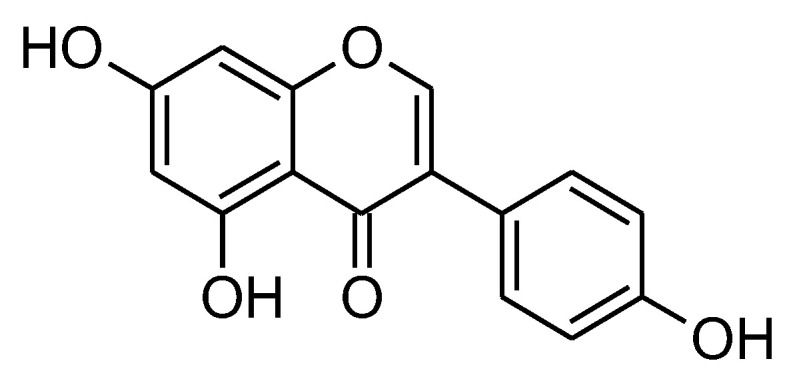
Chemical structure of genistein.

**Figure 4 molecules-27-05452-f004:**
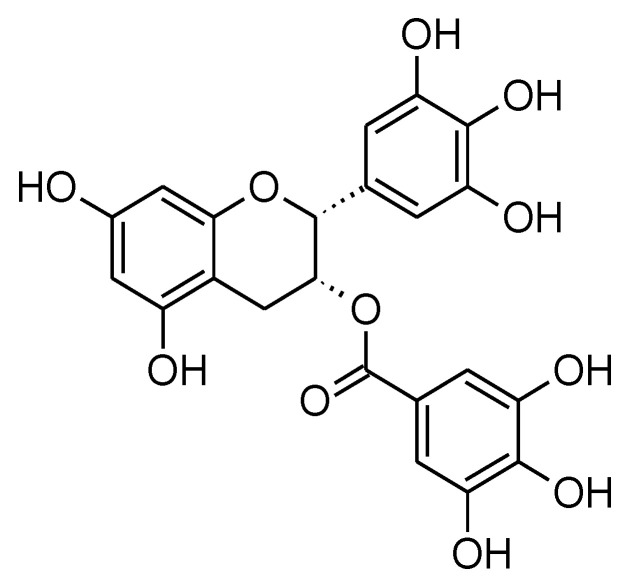
Chemical structure of EGCG.

**Figure 5 molecules-27-05452-f005:**
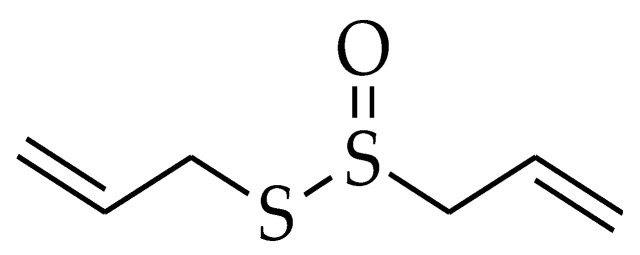
Chemical structure of allicin.

**Figure 6 molecules-27-05452-f006:**
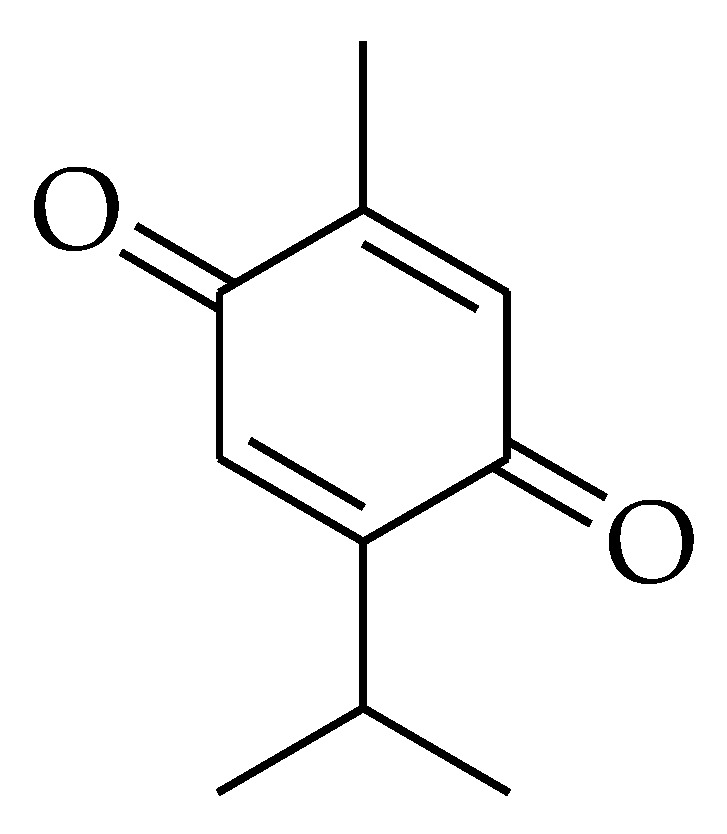
Chemical structure of thymoquinone.

**Figure 7 molecules-27-05452-f007:**
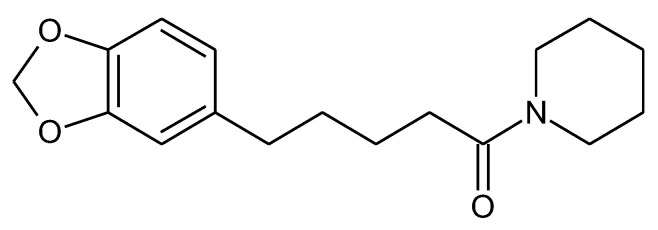
Chemical structure of piperine.

**Figure 8 molecules-27-05452-f008:**
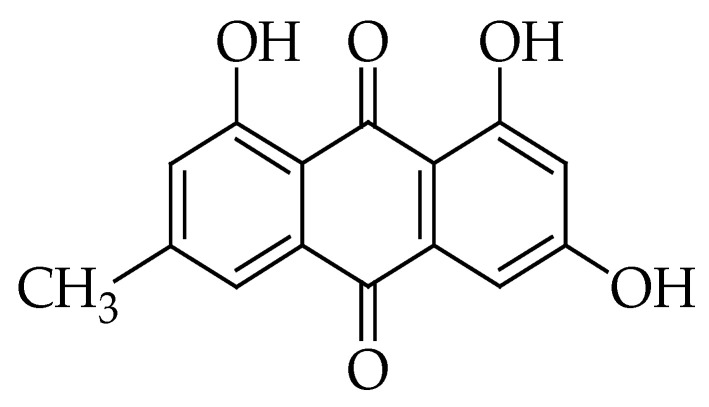
Chemical structure of emodin.

**Figure 9 molecules-27-05452-f009:**
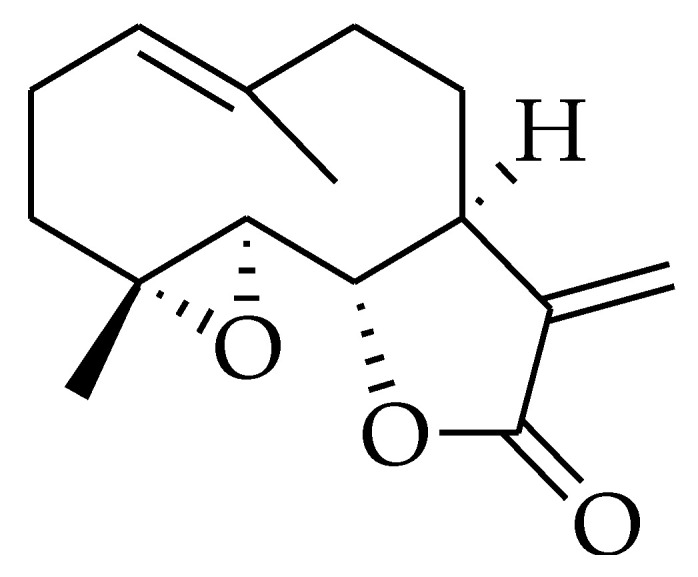
Chemical structure of parthenolide.

**Figure 10 molecules-27-05452-f010:**
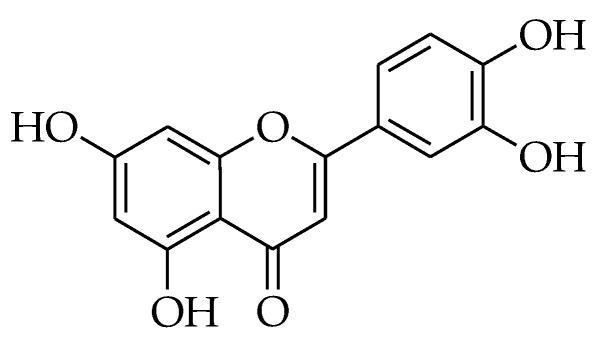
Chemical structure of luteolin.

**Figure 11 molecules-27-05452-f011:**
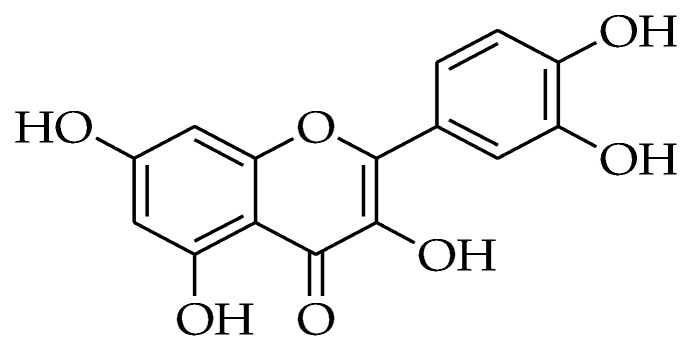
Chemical structure of quercetin.

**Figure 12 molecules-27-05452-f012:**
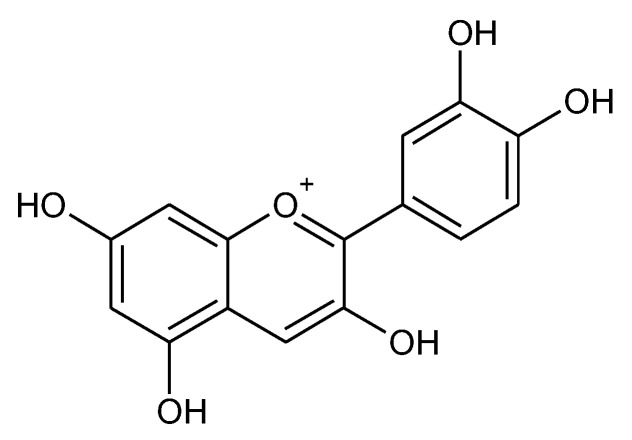
Chemical structure of anthocyanins (cyanidin).

**Figure 13 molecules-27-05452-f013:**
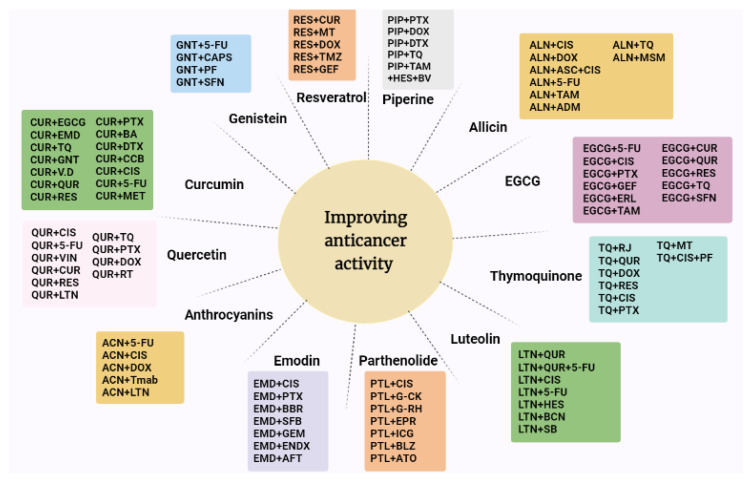
A summary of the natural compounds with their combination therapy. QUR, quercetin; CUR, curcumin; TQ, Thymoquinone; LTN, Luteolin; ACN, anthocyanins; PTL, parthenolide; GNT, genistein; PIP, piperine; EMD, emodin; RES, resveratrol; ALN, allicin; CIS, cisplatin; DOX, doxorubicin; MT, melatonin; TMZ, temozolomide; Tmab, trastuzumab; TAM, tamoxifen; DTX, docetaxel; PTX, paclitaxel; CCB, celecoxib; CAPS, capsaicin; PF, photofrin; SFN, sulforaphane; GEF, gefitinib; ASC, ascorbic acid; ADM, Adriamycin; MSM, methylsulfonylmethane; RJ, royal jelly; PF, pentoxifylline; BV, bee venom; HES, hesperidin; BBR, berberine; SFB, sorafenib; AFT, afatinib; GEM, gemcitabine; ENDX, endoxifen; G-CK, ginsenoside compound k; G-Rh, ginsenoside Rh; EPR, epirubicin; ICG, indocyanine green; ATO, arsenic trioxide; BLZ, balsalazide; SB, silibinin; BCN, baicalein; VIN, vincristine; RT, radiotherapy.

**Table 1 molecules-27-05452-t001:** Combination of experimental design of natural compounds with other natural products and the outcomes of these studies.

Natural Compounds	Chemical Classification	Combination Therapy	Concentrations Used	Type of Cancer	Experimental Model	Outcomes of the Combination	Intersecting Mechanisms	References
**Curcumin**	Diarylheptanoid, phenolic compound	Curcumin/Resveratrol	Curcumin 15 mM Resveratrol 15 μM	Breast cancer Salivary cancer	In vitro	Reducing cancer cell viability, increased ER stress and activation of the pro-death UPR protein CHOP	Apoptosis	[49]
		Curcumin/Soy isoflavones	Curcumin 20 mM Isoflavones 10 mg/mL	Prostate adenocarcinoma	In vitro	Reduced the concentration of PSA	Anti-androgen effect	[48]
		Curcumin/Emodin	Curcumin 30 μM Emodin 80 μM	Breast cancer	In vitro	Reduced tumor growth and invasion by inducing the expression of miR-34a	Inhibition of proliferation and invasion of breast cancer cells through upregulation of miR-34a	[55]
		Curcumin/ EGCG	Curcumin 3 mM EGCG 25 μM	Breast cancer	In vitro In vivo	Suppress ERα-breast cancer cell growth	G2/M-phase cell cycle arrest	[54]
		Curcumin/Thmoquinone	Curcumin 24.91 µM TQ 41.16 µM	Breast cancer	In vitro	Showed synergistic effect in reducing tumor cells growth via increasing caspase-3 and decrease PI3K and AKT	Cell proliferation inhibition Apoptosis induction	[56]
		Curcumin/Gemcitabine	Curcumin 10 μmol/L Gemcitabine 50 nmol/L	Pancreatic cancer	In vitro In vivo	Prevent the production, development, invasion, and metastasis of proteins (NF-B, EGFR, VEGF, COX-2, miRNA-22, Bcl-2, Bcl-xL, and others) upregulating Bax and caspases	Inhibition of proliferation, angiogenesis, and invasion	[58]
		Curcumin/Vitamin D	Curcumin 10^−5^ M 1.25D 10^−7^ M	Colon cancer	In vitro	Improved anticancer effect by interacting with vitamin D receptors	Activating vitamin D receptor (VDR) inducing the VDR target genes CYP3A4, CYP24, p21 and TRPV6. In the colon, some of these yet-to-be identified genes may play a role in cancer chemoprevention	[59]
		Curcumin/Quercetin	curcumin 3.1 μM and 6.2 μM Quercetin 25 μM and 50 μM	Human malignant melanoma	In vitro	Inhibition of proliferation, modulation of Wnt/β-catenin signaling and apoptotic pathway	Inhibition of cell proliferation through down-regulation of Wnt/β-catenin signaling pathway proteins, DVL2, β-catenin, cyclin D1, Cox2, and Axin2	[60]
		Curcumin/Boswellic acid	curcumin, 10 μmol/L AKBA 30 μmol/L	Colorectal cancer	In vitro In vivo	Induced chemoprevention through modulating miRNAs and their downstream target genes involved in cell-cycle control	Suppression of tumor growth by Induction the upregulation of tumor-suppressive miR-34a and downregulation of miR-27a in colorectal cancer cells	[47]
**Resveratrol**	Stilbeniod, phenolic compound, and a phytoalexin	Resveratrol/Curcumin	Resveratrol dose level of 5.7 mg/mL three times a week Curcumin dose level of 60 mg/kg of body weight three times a week	Lung cancer	In vivo	Synergistically stimulated p21 and modulated Cox-2 expression	expression of p21 significant decrease in tumor incidence and multiplicity curcumin and resveratrol have been reported to modulate p21 expression by a p53-dependen pathway adequate zinc levels along with phytochemicals resulted in efficient cell cycle arrest by p21 to control rapid cell proliferation	[80]
		Resveratrol/Melatonin	Resveratrol pellets in a concentration of 100 mg/kg Melatonin Drinking water pellets in a concentration of 100 mg/kg	Breast cancer	In vivo	NMU-induced mammary carcinogenesis was not affected by either agent alone, but when they were combined it resulted in a significant decrease in tumor incidence.	reduced tumor incidence by approximately 17% and significantly decreased the quantity of invasive and in-situ carcinomas returned food intake to the level of intact controls (significantly increased food intake) protective effects on NMU-induced rodent breast cancer	[81]
**Genistein**	Phytoestrogenic isoflavone	Genistein/Capsaicin	genistein 50 μmol/L Capsaicin 50 μmol/L	Breast cancer	In vitro	Synergistic apoptotic and anti-inflammatory effects	Reduced cell viability chromatin condensation and nuclear fragmentation stimulating AMPKα1	[97]
		Genistein/Sulforaphane	Genistein 15 µM Sulforaphane 5 µM	Breast cancer	In vitro	Promoted cell cycle arrest	downregulated KLF4 downregulated HDAC activity especially HDAC2 and HDAC3 downregulated hTERT	[101]
**EGCG**	Catechin/polyphenol	EGCG/curcumin	EGCG 50 and 100 μM curcumin 50 μM	Prostate cancer	In vitro	Arrested S and G2/M cycles	Arrested both S and G2/M phases of cell cycle Synergic up-regulation of p21 and followed cell growth arrest	[116]
		EGCG/Quercetin	EGCG 100 μM Quercetin 10 and 100 μM	Breast cancer	In vitro	EGCG had improved the anti-metabolic effect of quercetin in ER-negative breast cancers also it had decreased the viability and proliferation of MCF7 cells	Decreased cellular proliferation Inhibit glucose uptake by cells Metabolic antagonists in breast cancer cells, independently of estrogen signaling	[117]
		EGCG/Resveratrol	EGCG 30 μM resveratrol 15 μM	Head and neck cancer	In vivo	Enhanced apoptotic effect and reduced tumor growth	Increased apoptosis	[120]
		EGCG/Sulforaphane	EGCG 20 mM Sulforaphane 10 mM	Ovarian cancer	In vitro	Provoked apoptosis in ovarian resistant cells through human telomerase reverse transcriptase(hTERT) and Bcl-2 down regulation	arrest cells in both G2/M and S phase increases apoptosis in paclitaxel-resistant SKOV3TR-ip2 cells by down-regulating of hTERT and Bcl-2 and promote DNA damage response reducing the expression of hTERT	[119]
**Allicin**	Thiosulfinate	Allicin/ Thymoquinone	PC3 cells Allicin 24 g/mL Thymoquinone 500 g/mL CaCo_2_ cell Allicin 12 g/mL Thymoquinone 500 g/mL	Prostate and colon cancer	In vitro	Modulated antioxidant parameters	Increase of catalase activity in both PC3 cells and Caco2 cell	[141]
		Allicin/Methylsulfonylmethane	They used the IC50 MSM/allicin For CD44− 55.71 ± 8.47 mg/mL MSM/allicin For CD44+ 68.83 ± 9.78 mg/mL	Breast cancer	In vitro	Increased expression of caspase-3 mRNA expression	Enhanced more caspase-3 mRNA expression than allicin alone in both CD44± cells. Modulating the expression of the key apoptotic factors.	[143]
**Thymoquinone**	Monoterpenoid	Thymoquinone/Royal jelly	Thymoquinone 15 µmol/L Royal jelly 5 µg/mL	Breast cancer	In vitro	Enhanced anticancer activity	cell viability inhibition and PreG1 increase	[172]
		Thymoquinone/Quercetin	Thymoquinone 5 μM Quercetin 22.49 and 25.9 μM	Non-small cell lung cancer	In vitro	Induced apoptosis by modulating Bax/Bcl2 cascade	reduce the expression of antiapoptotic protein Bcl2 and induce proapoptotic Bax	[174]
		Thymoquinone/ferulic acid	Thymoquinone 50 and 100 µM ferulic acid 450 µM	Breast adenocarcinoma	In vitro	Synergic growth inhibition	decreased cell proliferation	[173]
		Thymoquinone/Melatonin	Thymoquinone 10 mg/kg/day Melatonin 1 mg/kg twice daily	Breast cancer	In vitro In vivo	Synergic antitumor effect by reducing tumor size with a 60% cure	induction of apoptosis, angiogenesis inhibition, and activation of T helper 1 anticancer immune response	[171]
		Thymoquinone/Resveratrol	TQ 46.03 μM Resveratrol 64.54 μM	Hepatocellular carcinoma	In vitro	Significant cell inhibition and increased caspase-3	cell inhibition and increase in caspase-3 indicating cell apoptosis raised reactive oxygen species leading to decrease of glutathione	[162]
**Piperine**	Alkaloids	Piperine/Thymoquinone	Piperine 425 μM Thymoquinone 80 μM	Breast cancer	In vivo	Inhibition of angiogenesis, induction of apoptosis, and shift toward T helper1 immune response	decrease VEGF expression and increased serum INF-γ levels angiogenesis inhibition, apoptosis induction, and shifting the immune response toward T helper1 response.	[181]
**Emodin**	Anthraquinonoe/phenolic compound	Emodin/berberine	Emodin 5–20 μM berberine 5–30 μM	Breast cancer	In vitro	Synergic inhibition of SIK3/mTOR pathway and induction of apoptosis	Attenuated aerobic glycolysis and cell growth as well as induce cell death by suppressing the SIK3/mTOR/Akt signaling pathway	[220]
**Parthenolide**	Sesquiterpene/germacranolide class	Parthenolide/ginsenoside compound k	parthenolide 7.5 mg/kg ginsenoside compound k 37.5 mg/kg	Lung cancer	In vitro In vivo	Increased tumor targeting	induce mitochondria-mediated lung cancer apoptosis	[233]
		Parthenolide/betulinic acid/honokiol/ginsenoside Rh2	Parthenolide 20.5 mg/kg, betulinic acid 20.3 mg/kg Honokiol 20.7 mg/kg ginsenoside Rh2 20 mg/kg	Lung cancer	In vitro In vivo	Displayed a synergistic activity in liposome systems for lung cancer treatment	cocktail liposome systems may provide a more efficient and safer treatment for lung cancer.	[234]
**Luteolin**	Digitoflavone/flavonoid	Luteolin/Baicalein	Luteolin 2.5, 5, 12.5, 25, 50, 80 and 100 mM Baicalein 2.5, 5, 12.5, 25, 50, 80 and 100 mM	Colorectal adenocarcinoma	In vitro	Synergic growth inhibition	inhibit cancer cells proliferation	[255]
			Luteolin 10 or 20 μM Quercetin 10, 20, and 40 μM	Cervical cancer	In vitro	Reduction in ubiquitin E2S expression led eventually to metastatic inhibition of cervical cancer	inhibited UBE2S expression	[247]
		Luteolin/Hesperidin	Hesperidin 100 μg/mL Luteolin 100 μg/mL	Breast cancer	In vitro	Induced cell cycle arrest by mediating apoptosis and downregulation the miR-21 expression	inhibition of cell proliferation, migration, and invasion reduced cell viability accumulation of apoptotic cells into the G0/G1 and sub-G1 cell cycle phases induced apoptosis through the intrinsic and extrinsic pathways, down-regulated anti-apoptotic, Bcl-2, and upregulated pro-apoptotic, Bax downregulated the expression of miR-21 and upregulated that of miR-16 and -34a in MCF-7	[249]
		Luteolin/Silibinin	Luteolin 20 µM Silibinin 50 µM	Glioblastoma	In vitro	Synergic inhibition of cell proliferation, migration, and invasion	inhibition of cell migration block angiogenesis block survival pathways leading to induction of apoptosis.	[247]
**Quercetin**	Flavonol/flavonoid	Quercetin/Curcumin	Quercetin 20 µM Curcumin 10 µM	Breast cancer	In vitro	Altered the BRCA1 deficiency and therefore augment the activity of anti-cancer drugs	synergistic action was observed in modulating the BRCA1 level and in inhibiting the cell survival and migration of TNBC cell lines	[258]
			Quercetin 11.39, 0.419 µM, Curcumin 2.85, 53.89 µM	Myeloid leukemia	In vitro	Enhanced apoptotic effect increasing ROS production	act indirectly on inhibition of STAT3 in a number of leukaemia cell lines (HL-60, U-937 and K562)	[259]
		Quercetin/Resveratrol	Quercetin 10 µM Resveratrol 10 µM	Oral cancer	In vitro	Cell growth inhibition, stimulation of apoptosis also it had been noticed to downregulate Histone deacetylase (HDAC)1, HDAC3, and HDAC8	Cell Growth Inhibition, DNA Damage, Cell Cycle Arrest, and Apoptosis in Oral Cancer Cells	[260]
			Quercetin 2 μg/mL Resveratrol 50 μg/mL	Skin cancer	In vivo Ex vivo	Synergistic effect over the use of single drugs	dual drug-loaded nanostructured lipid carrier (NLC) gel of quercetin and resveratrol enhanced their disposition in dermal and epidermal layers	[261]
		Quercetin/Thymoquinone	Quercetin 22.49 µM TQ 22.49 µM	Non-small lung cancer	In vitro	Downregulated BcL2, and activated BAX protein	reduce the expression of antiapoptotic protein Bcl2 and induce proapoptotic Bax, suggestive of sensitizing NSCLS cells toward apoptosis.	[174]
		Quercetin/Luteolin	Luteolin 10 or 20 μM Quercetin 10, 20, and 40 μM	Cervical cancer	In vitro	Lowered the ubiquitin E2S ligase (UBE2S) expression	inhibited UBE2S expression	[248]
**Anthocyanins**	Flavylium/flavonoid	Anthocyanins/luteolin	Anthocyanins Cyanidin-3-O-glucoside chloride 35 μmol/L luteolin 10 μmol/L	Breast cancer Colon cancer	In vitro	Increased apoptosis and inhibited proliferation	inhibited proliferation and increased apoptosis	[287]

**Table 2 molecules-27-05452-t002:** Combination experimental design of natural compounds with conventional anticancer therapy and the outcomes of these studies.

Natural Compound	Combination Therapy	Concentration Used	Type of Cancer	Outcomes of the Combination	Intersecting Mechanism	References
**Curcumin**	Curcumin/Paclitaxel	Curcumin 5 µM Taxol 5 nM	Cervical cancer	Curcumin enhanced paclitaxel-induced apoptosis by increasing p53 expression, activation of caspase-3, 7, 8, and 9, cleavage of poly(ADP-ribose) polymerase (PARP), and cytochrome c release	Non intersecting Curcumin enhanced paclitaxel-induced apoptosis by down-regulation of Nuclear Factor-κB and the Serine/Threonine Kinase Akt	[35,36]
	Curcumin/Docetaxel	Curcumin 20 μM Docetaxel 10 nM	Prostate cancer	Reduced docetaxel-induced drug resistance and side effects	Non intersecting curcumin enhances the efficacy of docetaxel treatment by inhibiting proliferation and inducing apoptosis through modulation of tumor-suppressor proteins, transcription factors and oncogenic protein kinases compared to each treatment alone	[38]
	Curcumin/Metformin	Curcumin 5–40 μM Metformin 0.4–12 mM	Prostate cancer	Synergistic impact on growth inhibition by apoptotic induction than curcumin and metformin alone	Apoptosis	[40]
	Curcumin/5-FU	curcumin 5 µM 5-FU 0.1 µM	Colorectal cancer	Overcome the drug resistance caused by 5-FU	Non-intersecting Curcumin decreases cancer stem cells and making cancer cells more sensitive to 5-FU	[42]
	Curcumin/Celecoxib	Curcumin 10–15 μmol/L Celecoxib 5 μmol/L	Colorectal cancer	Inhibited cancer cell proliferation	Growth inhibition was associated with inhibition of proliferation and induction of apoptosis. Curcumin augmented celecoxib inhibition of prostaglandin E2 synthesis. The drugs synergistically down-regulated COX-2 mRNA expression.	[43]
	Curcumin/Cisplatin	Curcumin 10 M Cisplatin 10 M	Bladder cancer	Stimulated caspase-3 and overexpression phospho-mitogen-activated protein kinase (p-MEK) and phospho-extracellular signal-regulated kinase 1/2 (p-ERK1/2) signaling	activating caspase-3 and upregulating phospho-mitogen-activated protein kinase (p-MEK) and phospho-extracellular signal-regulated kinase 1/2 (p-ERK1/2) signaling	[44]
	Curcumin/Doxorubicin	Curcumin 5 M Doxorubicin 0.4 mg/mL	Hodgkin lymphoma	Reduced cell growth by 79%	reduced cell growth by 79%, whereas each drug alone reduced L540 cell growth by 44% and 23%	[45]
**Resveratrol**	Resveratrol/Temozolomide	Resveratrol 12.5 mg/kg Temozolomide 10 mg/kg TMZ	Malignant glioma	Enhanced temozolomide’s therapeutic efficacy by inhibiting ROS/ERK-mediated autophagy and improving apoptosis	reduced tumor volumes by suppressing ROS/ERK-mediated autophagy and subsequently inducing apoptosis protected glioma cells from apoptosis, thus improving the efficacy of chemotherapy for brain tumors.	[78]
	Resveratrol/Doxorubicin	Resveratrol 25 µM Resveratrol 10–100 µM Resveratrol 12.5 mg/kg	Melanoma	Induced cell cycle disruption and apoptosis, resulting in decreased melanoma growth and increased mouse survival	Non intersecting resveratrol inhibits the growth of a doxorubicin-resistant B16 melanoma cell subline (B16/DOX) induced G1-phase arrest followed by the induction of apoptosis reduced the growth of an established B16/DOX melanoma and prolonged survival (32% compared to untreated mice).	[79]
**Genistein**	Genistein/5-FU	genistein 1.3 mg/day intraperitoneally FU 60 mg/kg, intraperitoneally	Pancreatic cancer	Tumor cells were augmented by the addition of genistein, which increased both apoptosis and autophagy	Non intersecting Genistein can potentiate the antitumor effect of 5-FU by inducing apoptotic as well as autophagic cell death.	[99]
	Genistein/Photofrin	genistein (0, 50, 100 μM) Photofrin (0–50 μg/mL)	Ovarian cancer Thyroid cancer	Enhanced the efficacy of photofrin-mediated photodynamic therapy	Non intersecting genistein sensitizes the activity of photodynamic therapy by photofrin in SK-OV-3 cells by inducing apoptosis through the activation of caspase-8 and caspase-3	[51]
	Genistein/Estradiol	Genistein 20 μM Estradiol 20 μM	Human liver cancer	Enhanced apoptosis	Enhanced apoptosis	[98]
**EGCG**	EGCG/5-FU	EGCG 50 μM 5-FU 10 μM	Colorectal cancer	Improved tumor cell’s sensitivity to 5-FU through inhibition of 78-kDa glucose-regulated protein (GRP78), NF-KB, miR-155-p5 and multidrug resistance mutation 1 (MDR1) pathways	Non intersecting EGCG enhanced the chemo-sensitivity of 5-FU in low doses by inhibiting cancer proliferation, promoting apoptosis and DNA damage EGCG blocked GRP78 expression, followed by enhancement of NF-κBand miR-155–5p level, which further inhibited the MDR1 expression and promoted the 5-FU accumulation in tumor cell	[87]
	EGCG/Cisplatin	EGCG 10 μM Cisplatin 10 μM	Ovarian cancer	Enhanced cisplatin sensitivity in ovarian cancer by regulating the expression of copper and cisplatin influx transport which is well-known as copper transporter 1 (CTR1)	DNA damage	[125]
	EGCG/Tamoxifen	EGCG 25 mg kg^−1^ Tamoxifen 75 μg kg^−1^	Breast cancer	Decreased the expression of EGFR, mTOR, and CYP1B	Decreased the expression of EGFR, mTOR, and CYP1B	[126]
	EGCG/Paclitaxel	EGCG 20 μM Paclitaxel 1 μM	Breast cancer	EGCG had synergistically encouraged the effect of paclitaxel by enhancing the phosphorylation of c-Jun N-terminal kinase (JNK)	induced 4T1 cells apoptosis	[127]
	EGCG/Gefitinib	EGCG 20 μM Gefitinib 1.25 μM	Non-small cell lung cancer	Inhibition of epithelial-Mesenchymal transition (EMT), and blocking of mTOR pathway	inhibit proliferation of HCC827-Gef cells	[128]
	EGCG/Erlotinib	EGCG 30 μM Erlotinib 1 μM	Head and neck cancer	enhanced apoptosis through the regulation of Bcl-2-like protein11(BIM) and B-cell lymphoma 2(Bcl-2)	inhibiting the phosphorylation of ERK and AKT and expression induces apoptosis of SCCHN cells by regulating Bim and Bcl-2 at the posttranscriptional level.	[129]
**Allicin**	Allicin/Cisplatin	Allicin 10 μg/mL Cisplatin 2 μg/mL	Lung cancer	Allicin overcome hypoxia mediated cisplatin resistance by increasing ROS production	shifts the mechanism of cell death towards more apoptosis allicin induced increase in ROS accumulation thus enhances cisplatin sensitivity even at low doses in A549 cells.	[144]
	Allicin/5-FU	Allicin 5 mg/kg/d; every two days for 3 weeks 5-FU 20 mg/kg/d 5 consecutive days	Hepatic cancer	Improved its sensitivity in hepatic cancer cells due to induction of apoptosis by ROS-mediated mitochondrial pathways	increased intracellular reactive oxygen species (ROS) level, reduced mitochondrial membrane potential (ΔΨm), activated caspase-3 and PARP, and down-regulated Bcl-2	[154]
	Allicin/Adriamycin	Allicin 25 μg/mL Adriamycin 2.5 μg/mL	Gastric cancer	Inhibited the proliferation and induced apoptosis	induced apoptosis and inhibited proliferation	[148]
	Allicin/Tamoxifen	Allicin 10 nM Tamoxifen 1 μM	Breast cancer	Improved the effectiveness of tamoxifen	Non intersecting Allicin in MCF-7 cells enhances the effectiveness of tamoxifen in the presence and absence of 17-b estradiol	[149]
**Thymoquinone**	Thymoquinone/Doxorubicin	For most experiments Thymoquinone 10 µM TQ Doxorubicin 50 nM for 24 h for the treatment of HuT102 cells for 48 h Thymoquinone 40 µM Doxorubicin 100 nM	Adult T-cell leukemia	Increased ROS production resulting in disruption of the mitochondrial membrane	Increased ROS production resulting in disruption of the mitochondrial membrane inhibition of cell viability and increased sub-G1 cells reduced tumor volume	[169]
	Thymoquinone/Cisplatin	Thymoquinone 20 mg·kg^−1^ oral cisplatin 2 mg·kg^−1^ ip	Hepatocellular carcinoma	Improved the effectiveness of Cisplatin via controlling the GRP78/CHOP/caspase-3 pathway	reduced the elevated GRP78 and induced CHOP-mediated apoptosis in the diseased liver tissues normalized alpha-fetoprotein (AFP) levels and improved liver functions	[167]
	Thymoquinone/Cisplatin/Pentoxifyllin	Thymoquinone i.p. (20 mg/kg) Cisplatin 7.5 mg/kg twice Pentoxifyllin s.c. route 15 mg/kg	Breast carcinoma	Enhance the effect of the treatment by Notch pathway suppression	reduced Notch1, Hes1, Jagged1, β-catenin, TNF-α, IL-6, IFN-γ, and VEGF with increment in IL-2, CD4, CD8, and apoptotic cells Notch suppression.	[170]
	Thymoquinone/Paclitaxel	100:1 μM of TQ with PTX	Breast cancer	increased the rate of apoptotic/necrotic cell death	Non intersecting Thymoquinone does not improve Paclitaxel potency against MCF-7 or T47D cells and apparently antagonizes its killing effects. However, TQ significantly abolishes tumor-associated resistant cell clones Thymoquinone enhanced Paclitaxel induced cell death including autophagy TQ significantly increased the percent of apoptotic/necrotic cell death in T47D cells after combination with paclitaxel induced a significant increase in the S-phase cell population	[168]
**Piperine**	Piperine/Paclitaxel	5:1	Breast cancer	Synergistic anticancer effect	Non intersecting piperine can improve the bioavailability of paclitaxel and can potentiate the antitumor effect of paclitaxel	[189]
	Piperine/hesperidin/bee venom/Tamoxifen	Piperine 34.89 μg/mL Hesperidin 12.14 μg/mL bee venom 10.19 μg/mL Tamoxifen 2.98 μg/mL	Breast cancer	Enhance the anti-cancer effects of tamoxifen	Enhance the anti-cancer effects of tamoxifen	[190]
	Piperine/Doxorubicin	Piperine 50 µM Doxorubicin 10 µM	Breast cancer	Inhibited tumor growth	Piperine enhanced the cytotoxicity effect of doxorubicin	[191]
	Piperine/Docetaxel	Piperine 50 mg/kg p.o. Docetaxel 12.5 mg/kg	Prostate cancer	Improved the antitumor efficacy of docetaxel	Improved Anti-Tumor Efficacy Via Inhibition of CYP3A4 Activity	[192]
**Emodin**	Emodin/Sorafenib	Emodin 20 μM Sorafenib 0.5 μM and 1 μM	Hepatocellular carcinoma	Improving the anti-cancer effect of sorafenib by increasing apoptosis and cell cycle arrest	Non intersecting emodin synergistically increased cell cycle arrest in the G1 phase and apoptotic cells in the presence of sorafenib	[207]
	Emodin/Afatinib	Emodin 50 mg/kg/day for 4 weeks Afatinib 50 mg/kg/day for 4 weeks;	Pancreatic cancer	Inhibited cell proliferation	Regulating the Stat3 expression.	[216]
	Emodin/Cisplatin	Emodin A549 cells:5 µM H460 cells, 2.5 µM Cisplatin A549: 8, 10 and 15 µM H460 cells:2, 4, 6, 8 and 10 µM	Lung adenocarcinoma	Increased cisplatin sensitivity through P-glycoprotein downregulation	Non intersecting Emodin inhibited the proliferation of A549 and H460 cells emodin enhanced cisplatin-induced apoptosis and DNA damage in A549 and H460 cells emodin can increase A549 and H460 cell sensitivity to cisplatin by inhibiting Pgp expression	[219]
	Emodin/Paclitaxel	Emodin 10 μM Paclitaxel 4 μM	Non-small cell lung cancer	Enhanced the antiproliferative effect of paclitaxel	Inhibited the proliferation of A549 cells	[212]
	Emodin/Gemcitabin	Emodin 40 μM Gemcitabine 20 μM	Pancreatic cancer	Emodin inhibited IKKβ/NF-κB signaling pathway and reverses Gemcitabine resistance	Increase the apoptosis rate	[213]
	Emodin/Endoxifen	Emodin 0, 15, 30, 60 µM Endoxifen 0, 2, 4 µM	Breast cancer	Elevation of cyclin D1 and phosphorylated extracellular signal-regulated kinase (pERK)	Emodin attenuated tamoxifen’s treatment effect via cyclin D1 and pERK up-regulation in ER-positive breast cancer cell lines.	[294,299]
**Parthenolide**	Parthenolide/Epirubicin	Parthenolide 2.5, 0.75 and 0.2 µM Epirubicin (9, 7, and 5 µM	Breast cancer	improved cytotoxicity and apoptosis as well as reduced the undesirable side effects	Up-regulated the expression of Bax as a pro-apoptotic gene in MDA-MB cells down-regulated the expression of Bcl2 as an anti-apoptotic gene in MDA-MB cells increasing the fracture of caspase 3 and improving the apoptosis pathway	[221]
	Parthenolide/Indocyanine		Breast cancer	Synergistic antitumor activity	More ROS-mediated killing of the tumor cells by exerting a synergistic effect for treating triple-negative breast cancer	[270]
	Parthenolide/Arsenic trioxide	Parthenolide 1 μg/mL Arsenic trioxide 2 µM	Adult T-cell leukemia/lymphoma	Enhanced the activity	Non intersecting parthenolide significantly enhanced the toxicity of ATO in MT2 cells.	[231]
	Parthenolide/Balsalazide	Parthenolide 5 and 10 μmol/L Balsalazide 20 mmol/L	Colorectal cancer	Improved the anticancer activity via blocking NF-κB activation	Exhibits synergistic suppression of NF-κB and NF-κB–regulated gene products that are associated with apoptosis, proliferation, invasion, angiogenesis, and inflammation	[232]
**Luteolin**	Luteolin/Cisplatin	Luteolin 0, 10, 50, 100 μM Cisplatin 2 μg/mL	Ovarian cancer	Significantly sensitized the antineoplastic effect of cisplatin by initiating apoptosis and inhibiting cell invasion and migration	Suppressing CAOV3/DDP cell growth and metastasis inducing apoptosis by decreasing Bcl-2 expression.	[245]
	Luteolin/5-FU	Luteolin:5-fluorouracil 10:1, 20:1, 40:1 luteolin:100, 50, 25, 12.5, 6.25, 3.125 µM 5-FU: 10, 5, 2.5, 1.25, 0.5, 0.25 µg/mL	Hepatocellular carcinoma	synergistic anticancer effect	Apoptosis induction and metabolism	[244]
**Quercetin**	Quercetin/Cisplatin	Quercetin 100 μM cisplatin 5 μg/mL	Oral squamous cell carcinoma	Inhibition of NF-κB thus downregulating of X-linked inhibitor of apoptosis protein(xIAP)	Induced apoptosis in human OSCC (cell lines Tca-8113 and SCC-15) by down-regulating NF-κB	[273]
		Quercetin 50 μM cisplatin 10 μM	Hepatocellular carcinoma	potentiated the growth suppression effect of cisplatin	Inducing growth suppression and apoptosis in HepG2 cells	[268]
		quercetin 15 μM cisplatin 10 μM	Cervical cancer	Induced apoptosis by downregulation of MMP2, METTL3, P-Gp and ezrin production	Promoting apoptosis and inhibiting proliferation, migration and invasion of cervical cancer cells	[262]
	Quercetin/Tamoxifen	Quercetin 50 μM Tamoxifen 10–6 mol/L	Breast cancer	Enhanced the activity	Proliferation inhibition and apoptosis in MCF-7Ca/TAM-R cells	[264]
	Quercetin/Vincristine	Vincristine 50 mg Quercetin 50 mg	Lymphoma	Potentiated the effect of vincristine	Synergistic effect through lipid-polymeric nanocarriers (LPNs) for the lymphoma combination chemotherapy	[269]
	Quercetin/Doxorubicin	Quercetin 0.7 μM Doxorubicin 2 μg/mL	Breast cancer	Suppression of efflux receptors (BCRP, P-gp, MRP1), and reduced the side effects of doxorubicin	Down-regulating the expression of efflux ABC transporters including P-gp, BCRP and MRP1 and attenuating the toxic side effects of high dose doxorubicin to non-tumor cells	[265]
		Quercetin and Doxorubicin 5 mg/kg	Gastric cancer	Improved the efficacy	Improved the efficacy of gastric carcinoma chemotherapy	[267]
		Doxorubicin 0.75 μM Quercetin 230 μM	Breast cancer	Improved the efficacy	Induction of apoptosis in cancer cells	[266]
	Quercetin/Radiotherapy	Theranostic system (CQM ) 50 μm	Breast cancer	Improved the tumor targeting and radiotherapy treatment	Promoted tumor cell apoptosis	[272]
	Quercetin/Paclitaxel	Quercetin 20 µM Paclitaxel 5 nM	Prostate cancer	Improved efficacy by by ROS production, induction of apoptosis, preventing cell migration and causing cell arrest in G2/M phase	Induction of apoptosis cell arrest in G2/M phase ROS production Preventing cell migration	[270]
		Quercetin 2, 10, 20 mg/kg Paclitaxel 40 mg/kg	Breast cancer	had enhanced the multi-drug resistance in breast cancer by decreasing P-gp expression	Lower IC50 value, higher apoptosis rate, obvious G2M phase arrest as well as stronger microtubule destruction in MCF-7/ADR cells	[271]
**Anthocyanins**	Anthocyanins/ 5-FU	Caco2 cells BRB Anthocyanins 50 μg/mL 5-FU 25 μM or 50 μM SW480 cells BRB Anthocyanins 50 μg/mL 5-FU 16 μM or 32 μM	Colorectal cancer	decreased the proliferation and migration of tumor cells	Decreased number of tumors decreased the proliferation	[287]
	Anthocyanins/Cisplatin	AIMs Anthocyanins 400 µg/mL Cisplatin 5 μg/mL	Breast cancer	advanced the sensitivity of cisplatin by inhibiting Akt and NF-κB activity	Non intersecting Anthocyanins isolated from Vitis coignetiae Pulliat (Meoru in Korea) (AIMs) Enhances Cisplatin Sensitivity in MCF-7 Human Breast Cancer Cells through Inhibition of Akt and NF-κB Activation	[289]
	Anthocyanins/Doxorubicin	Anthocyanins 1–25 μg/mL Doxorubicin 5 μM	Breast cancer	decreased doxorubicin cardiac toxicity	Smoothies containing mixtures of Citrus sinensis and Vitis vinifera L. cv. Aglianico N, two typical fruits of the Mediterranean diet decreased doxorubicin cardiac toxicity	[291]
	Anthocyanins/Trastuzumab	C3G 5 μg/mL Trastuzumab 5 μg/mL	Breast cancer	Improved trastuzumab apoptotic effect	Non intersecting Improved trastuzumab apoptotic effect	[294]
		C3G (1 mg/mL) or P3G (1 mg/mL)	Breast cancer	Overcome trastuzumab-resistant cells due to the decrease in HER2, AKT and MAPK activities	Non intersecting Anthocyanin overcome trastuzumab-resistant cells due to the decrease in HER2, AKT and MAPK activities inhibits invasion and migration of trastuzumab-resistant human breast cancer cells	[295]

## Data Availability

Not applicable.

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
