# Peer review of "Combination Anticancer Therapies Using Selected Phytochemicals"

_molecules, 2022, doi:10.3390/molecules27175452_

Round 1

Reviewer 1 Report

The review paper by Talib et al. describes the studies on natural products of terrestrial origin as anticancer agents when used in combination with chemotherapeutic agents and/or other natural products.

Although this review covers a large amount of the literature in this field, I personally found it confusing to follow at times and I was missing the reasoning behind this review paper. I would have liked to see an explanation, reasoning either based on literature data or personal opinion of the authors’ team on why combinatorial anticancer therapy is gaining attention. Although some of the reasons are expressed scatteredly in the manuscript, I believe that having this information in a more structured way at the beginning of the manuscript, such as in the introduction, will set the tone, the aims and the purpose of this review. It is important to include toxicity information of each NP that is mentioned as part of the combinatorial strategy.

In general, I found that there was lack of structure in some parts and I think that subheadings are needed within each section to make it clear when the authors are describing NP combination and when they refer to NP-chemo agents combination. Moreover, there seems to be random capitalisation of compound names (e g,. line 40, 86 ) and words (eg. Line 111, 116). Please check and correct throughout.

The names of the plants and the “in vitro” and “in vivo” are not in italics and that should also be corrected throughout, including the tables

In section 2.1, there are a lot of acronyms that are not examined whereas such thing does not happen in other sections. It needs to be kept consistent throughout the text and acronyms should be introduced and expanded at the beginning. In relation to that, the legend of Figure 13 is showing all the abbreviations used for the natural products but that should come earlier in the text.

The structure of the compounds also needs adjustments/corrections. For example, Figures 1, 3 are distorted, so please fix this. Also, for the structure of resveratrol, both cis and trans can be shown since they ae both mentioned in the text. The structure of EGCG needs to be cleaned up, especially for the -OH groups. Also, there is no need to show the stereochemistry of the hydrogen. Figure 12 is wrong, the -CH3 groups should be -R groups.

The manuscript needs to be thoroughly checked for English language as there are a lot of points that the sentences do not make sense and there are a lot of grammar mistakes that need to be corrected. There is inconsistency in the past tense used throughout the text and that needs to be corrected. The sentences are not linked appropriately, making the text a lot harder to flow smoothly. For example, in line 545 “on the other hand” suggested something contrary to what was described before, but that is not the case here.

In line 618 when Figure 13 is introduced that should be a separate paragraph and not a continuation of section 2.12. I believe that this would stand better in conclusions along with the tables. In line 645-646 it is mentioned that “other natural products were less tested”, This statement should be expanded and explain/suggest why it that. Moreover, the paragraph of the conclusions section is mentioning about further studies. It would add to the value of the paper if the authors can suggest/propose ideas/strategies for such further studies.

A lot, if not most of the references are missing the journal name. Please check

Below there are some more section-specific comments

I found section 2.1 hard to follow, there was a lot of information provided by not a clear way of showing the mechanisms of action like it is done in other sections. Especially when giving information based on clinical trails there needs to be more detail. A good example is lines 140-142. The same style should be followed throughout.

In Section 2.2 please change 40 to 4' in the name of resveratrol. What does it mean “newly synthesise”? I think the authors mean that this compound is synthesised as a response but that does not mean is newly synthesised. Why is the study of Malhorta et al (line 166) in this section and not in 2.1? I understand that there will be cases that will overlap but is not clear how this is decided.

In section 2.3 Ref 71 is described as “published just recently”. The paper was published in 2007 which makes it 15 years old! In line 207 it is mentioned that “other effects are not related to this activity”. Although there is a reference for that sentence, there is no information provided. What other effects? This is one of the many examples throughout the manuscript where there is a sentence with no real information but with a citation. Adding a reference for the sake of having 300 references but not introducing the study, does not add value to the paper. In line 222 it is mentioned “Resulted in the greatest apoptosis”: how great? This is very vague. Again, that is one of the many examples in the text where words such as great are used but are not giving a real value of the effect

In section 2.6 lines 349-360 should not be in italics

In section 2.7 please avoid starting all the sentences with the word “piperine”

In section 2.8 line 403 delete “chemically it is” In line 429-430, please explain why FASN is an important factor. In line 436 change to “…emodin can improve the anticancer…”. In line 438 sorafenib is introduced. Please add information on this anticancer agent such as for what types of cancer is used etc

Please change section 3.9 to 2.9. The sentence in lines 469-470 is out of place as the combination of NPs has already been mentioned as a strategy earlier in the text. Line 482-483, this sentence needs to be re-structured to make sense.

Section 2.10: Line 511 the verb combined refers to which compound? Luteolin?

Section 2.11: Line 529 change the word mechanisms to biological activities

Section 2.12: It is not clear if in each study that is described they are using a mixture of anthocyanins. If so, which ones? The composition of the mixture will affect the activity. Does the plant origin of the anthocyanins affect the activity? That can e compared only if the mixture has the same components. Line 596 please correct name of the plant

Author Response

Many thanks for your constructive comments and suggestions. All your comments were considered in the revised manuscript. Attached is a detailed response.

We hope that our revised manuscript will meet your expectations

Reviewer 2 Report

Dear Editor,

It was a pleasure for me to review the manuscript titled "Plants Derived Natural Products in Combination Anticancer Therapies" submitted by Talib et al.

Although the topic is of high interest, the presentation of the study is rather suboptimal. In addition, the level of the English language throughout the manuscript is very low.

Please find below my specific comments.

Title

It is too general and very broad. The authors present a few potential compounds that can be used in cancer.

Abstract

"... highlighted the different combinations of anticancer therapies using plant-derived natural products in combination with other chemotherapeutics agents such as drugs and plants...": are plants not also natural products? This sentence sounds redundant.

I think the section is very short. The authors should provide more details to fully orient the reader.

Introduction

Have a native English-speaking scientist review the section: "Everyone thinks that it is away from them but unfortunately," is not appropriate and is very unclear.

I find the section too general. Some examples of current drugs could be given with their limitations. Or even some examples of natural products that have shown efficacy in cancer treatment

There are multiple expressions that are not understood by a person of a different culture. The text should be written in standard English

Section 2: " Natural products in combination anticancer therapies"

Again this is too broad for a title. There are a great amount of other compounds not mentioned here.

The authors could group the 12 compounds based on their mechanisms of action. This would provide a clear rational why they can be combined with given regimens.

The report should be writing in standard English language.

Conclusion

A discussion section could be added, including perspectives and limitations of current combinations.

Author Response

Many thanks for your positive feedback and constructive comments. We revised the manuscript based on your comments and suggestions and attached a detailed response.

We hope that our modified manuscript will meet your expectations.

Reviewer 3 Report

The manuscript presents a number of plant-derived compounds with applications in anti-cancer therapy. The first part of the manuscript presents their effects both alone and in combination with other natural products. The interaction of these compounds with drugs used in anticancer therapy is also discussed. The second part of the manuscript includes Tables, which compile the most important information on the topic under discussion.

The manuscript is interesting, and contains a lot of useful information on the possibility of using compounds of natural origin as means of supporting the body as well as enhancing the effects of drugs.

Minor remarks

Figures need to be reduced to fit the text. In addition, Figures 1, 3, 12 are deformed, they should be corrected.

Item 2.2 - should be "Resveratrol (trans-3,5,4'-trihydroxystilbene).

Figure 2 shows the cis form of this compound, and it was supposed to be the trans form.

The authors should carefully check when it is required to write certain names with a capital letter, and when a lowercase letter is sufficient. This is because inconsistency is evident, e.g. Vinblastine, Vincristine, Taxol, Etoposide (lines 40-41) or vincristine, irinotecan, paclitaxel, and etoposide (lines 53-54). Similarly, on line 86 is Curcumin, and on line 88 is Curcumin. Later in the text there are more such names spelled unnecessarily with a capital letter.

In turn, in lines 102, 111 a new sentence begins with a lowercase letter instead of a capital letter.

Another comment concerns the way the abbreviation NF-κB is written, which is sometimes spelled as NF-B, NF-KB, NF-κβ, NF-k κβ, which needs to be corrected.

The sentence in lines 39-41 is repeated in part in lines 53-54. This needs to be corrected.

In line 318, 17-b estradiol appears, and I think it should be 17-beta.

In line 325 the fragment 4+ is underlined (unnecessarily), I think.

The text in lines 349-361 is badly formatted (italic font, left alignment).

The paragraph in lines 368-378 needs rewording. Starting each sentence with the word "piperine" sounds bad.

In line 403 in the sentence "Chemically it is (1,3,8-trihydroxy-6-methyl-anthraquinone)" the parenthesis is unnecessary.

In line 567-568, instead of improvemnet, it should be improvement.

Paragraph 2.12 and onward - should be "anthrocyanins" or "anthocyanins," as the two forms are used interchangeably.

Author Response

Thank you very much for your positive feedback and constructive comments.

The manuscript was revised based on your comments and suggestions.

We hope that the revised manuscript will meet your expectations.

Reviewer 4 Report

In this version, the review cannot be accepted. The main purpose of the reviews is not only to collect material from all available sources, but to rework and analyze it for the convenience of readers. Only then the review is in demand and cited. This review clearly does not analyze the material. To begin with, the choice of compounds is not explained at all. There are many natural compounds that exhibit anticancer activity and modify the action of other substances, including chemotherapeutics. It would be much better to limit ourselves to some types of substances (flavonoids or terpenoids or meroterpenoids or ...) and compare the mechanisms of enhancing anticancer activity within a group of compounds of the same structural type. Perhaps this would make it possible to isolate the main signaling pathways or predict similar activity for compounds with similar structures.

The review is severely lacking in quantitative data. It is not known at what doses these substances exhibit antitumor activity. In fact, almost any substance exhibits a cytotoxic / antitumor effect, the question of dosing. And in what doses (concentrations) they are used to enhance antitumor activity. Is it non-toxic? Be sure to quantify antitumor activity compared with activity against non-tumor cell lines. No analysis - the same mechanisms of action in therapy with an individual substance and in combination therapy.

The order of the chapters is strange. For example, Genistein is at the beginning of the review, and its closest relatives Luteolin and Quercetin are at the end. Although these compounds differ only in the arrangement of hydroxyl groups, one would expect them to act similarly. The absence or presence of such could be the subject of discussion.

The order within each chapter is also constantly violated. It would be logical to first give data on the action with cytostatics, and then with natural compounds. In half of the chapters, this is true, but in the rest, the order is either reversed, or the data is mixed within one paragraph. By the way, due to the fact that the data on the combined action of two natural substances in the case when they are both mentioned in the review intersect and it is not clear for what reasons they are spaced between chapters, then, in my opinion, it would be better to take out the data on the combined action of two natural substances from different chapters into one separate chapter. And it is possible to create a table with cross influence.

Another shortcoming of the review is the inaccurate presentation of data, from which it is often not clear what type of experiments were carried out: in vitro or in vivo. After clarification, the data should be sorted into different paragraphs, placing data from animal experiments below cell experiments.

The positive aspects of this work include the presence of tables 1 and 2 and Fig. 13, summarizing the given data. However, the text of review is not so good. Table 2 should also specify the type of experiment: in vitro or in vivo.

In fact, this review of new knowledge, except for those that can be found independently by searching for keywords, does not give. If we consider the review as a help for those who have limited access to literary sources, then there is not enough quantitative data for these purposes.

Author Response

(The authors gave the same response as above.)

Round 2

Reviewer 1 Report

The authors have taken into consideration edits and suggestions I have made to the previous version of their manuscript. They have put a great effort into re-structuring some parts of the review, adding more details (such as doses) when presenting studies and writing a more thorough conclusion and more inclusive introduction. Therefore, I would like to thank them for their effort, dedication and hard work.

This is an important piece of work and I believe that the English needs to be checked one more time (native English speaker would be preferable) before publishing it. Some examples where the English needs to be correct are the following. However, please note that they are not the only ones.

L18: consisting of natural products

L27: Remove “on the other hand” This sentence is still presenting negative facts associated with cancer

L32: not correct English

L90: not correct English

L154: Resveratrol has

L65: plant name not in Italics

Figure 12: what are the R groups? Maybe show the R groups of the anthocyanins mentioned in the section.

Author Response

Thank you very much for your comments and suggestions that helped us to improve our manuscript.

All your comments were considered in the revised manuscript and we hope that this version of the manuscript will meet your expectations.

Reviewer 2 Report

The authors have addressed most of my comments. However, the manuscript does not read well and requires extensive revision, particularly regarding the use of the English language.

Author Response

Thank you very much for your comments.

We did an extensive revision to improve English language and we hope that the revised manuscript will meet your expectations.

Reviewer 4 Report

The second version of the manuscript is significantly better than the first, but still has a number of significant shortcomings. They are primarily concerned with the organization of the manuscript.

I’ll show on the example of one of the first chapters 2.1 curcumin (in fairness, it should be noted that in other chapters the manuscript is better organized):

Lines 83-99 synergy with natural products

100-104 co-action with cytostatic gemcitabine

104-106 generally just about the mechanism of action

106-108 again combined action with natural products (quercetin)

109-117 co-action with paclitaxel, although of plant origin, but known cytostatic, in contrast to quercetin and other natural products

117 and beyond - again cytostatics

And at the end suddenly LD50, which should be given above, where cytotoxic concentrations are given.

By the way, LD50 is measured in mg/kg, not µg/mL. And the original data is correct.

The result is a poorly perceived mishmash.

I strongly recommend that the text in this chapter and in the following ones be arranged more logically.

First IC50 / CC50, then LD50 (it is better to give several data on different types of administration, or be sure to indicate the type of administration). It is important to place these indicators side by side in order to understand the severity of the intrinsic antitumor effect of the title compound. Then the mechanisms of its own antitumor action. Then co-action with cytostatics (including mechanisms), then with cytostatics of plant origin, and only after with natural compounds not used in antitumor therapy.

Author Response

Many thanks for you comments and suggestions. We did an extensive language revision to correct any language error. We also rearranged the details in section 2 as requested to include LD50 and IC50 values next to each other. Additionally, we added new details on therapeutic concentrations used in vivo to complete the information in this part.

We thank you again for helping us to improve our article and we hope that this version of the manuscript will meet your expectations.